# Understanding SAM through Minimax Perspective

Ying Chen [1]  Aoxi Li [1]  Javad Lavaei [1]

## Abstract

Sharpness-Aware Minimization (SAM) empirically boosts generalization by seeking parameters that minimize the worst-case loss in a small neighborhood, yet existing theory explains its behavior under either Polyak-Lojasiewicz (PL) condition or upper bounded perturbation radius. We revisit SAM through the bilevel minimax problem $\min_\theta \max_{\|\Delta\| \le \rho} l(\theta + \Delta)$ and derive a $(\theta, \Delta)$ gradient flow ODE whose equilibria coincide with the problem's optimality conditions. A Lyapunov argument-free of convexity assumptions, quantifies how the optimality gap depends on the radius $\rho$ and local curvature. Discretizing the flow yields a *Multi-step SAM* algorithm that recovers classical SAM as $\rho \to 0$. Moreover, our analysis and the resulting algorithm remain valid even for large $\rho$, providing guidance for aggressive neighborhood exploration. Experiments on synthetic objectives and CIFAR-10 validate the predicted gains from multiple inner updates, bridging the gap between SAM's minimax intuition and its practical implementation.

## 1. Introduction

Modern deep networks often generalize well despite being heavily over-parameterized, a behavior that has been empirically tied to the *flatness* (low sharpness) of the loss landscape at the learned parameters. Sharpness-Aware Minimization (SAM) turns this insight into practice by solving, at each update, a *min-max* sub-problem that minimizes the loss in the worst-case weight neighborhood (Foret et al., 2021). Although SAM and its variants now underpin state-of-the-art performance across vision, language and domain-generalization tasks (Wang et al., 2023; Zhang et al., 2024), their theoretical understanding remains highly unclear in the nonconvex setting with large $\rho$. Existing convergence

[1] Department of Industrial Engineering and Operations Research, University of Califorlia, Berkeley, California, United States. Correspondence to: Ying Chen <ying-chen@berkeley.edu>.

*Proceedings of the $43^{rd}$ International Conference on Machine Learning*, Seoul, South Korea. PMLR 306, 2026. Copyright 2026 by the author(s).

analyses to optimal solution either assume *convexity* of the loss (Luong et al., 2024) or require that the perturbation radius $\rho$ be *vanishingly small*, so that the SAM step reduces to first-order gradient descent (Andriushchenko & Flammarion, 2022; Shi et al., 2022). Otherwise even with PL condition, $\rho = \mathcal{O}(1)$ does not fully converge to the minimum (Oikonomou & Loizou, 2025; Dai et al., 2023) and there is constant level optimization error. Consequently, there is still no principled answer to a basic question, for a general non-convex function, when $\rho$ is not bounded, how will SAM perform, and

> *How far is the practical SAM update away from the ideal minimax objective that motivates it?*

### 1.1. Motivation and Problem Formulation

While the theoretical link between flatness and generalization remains a subject of active debate, complicated by re-parameterization invariance and scale dependence (Dinh et al., 2017; Andriushchenko et al., 2023), a substantial body of empirical evidence suggests that, for practical neural architectures, seeking regions of low curvature consistently correlates with improved test performance (Keskar et al., 2017; Foret et al., 2021). Normalization layers further complicate this landscape by driving optimization to an 'edge-of-stability' regime (Lyu et al., 2022), suggesting that the interplay between sharpness and dynamics is more critical than static geometric measures alone.

Sharpness-Aware Minimization (SAM) operationalizes these insights by solving the local min-max problem. Motivated by the original goal of SAM which minimizes the loss function at the worst direction perturbation, we focus on the specific bilevel-optimization problem $\min_\theta \max_{\|\Delta\|_2 \le \rho} L(\theta + \Delta)$, or more precisely,

$$\min_\theta L(\theta) := l(\theta + \Delta^*(\theta)), \tag{1a}$$

$$\text{s.t. } \Delta^*(\theta) \in \arg \max_{\|\Delta\|_2 \le \rho} l(\theta + \Delta) \tag{1b}$$

where $\theta \in \mathbb{R}^d$ represents the decision parameters and $\Delta \in \mathbb{R}^d$ is a perturbation bounded by the radius $\rho$. We treat the adversarial perturbation $\Delta$ as a latent variable and analyze the coupled dynamics of the outer *learner* $\theta$ and the inner *adversary* $\Delta$.

To capture the min-max objective, we focus exclusively on the normalized formulation of SAM. Existing frameworks for this variant typically require the perturbation radius $\rho$ to be either restricted (Oikonomou & Loizou, 2025) or vanishingly small (Khanh et al., 2024). While these assumptions simplify analysis by effectively linearizing the dynamics, they fail to capture the distinct behavior of the practical SAM, where $\rho$ is often large, constant, and interacts nonlinearly with the landscape curvature.

Recent works have begun to address this gap by analyzing SAM without the vanishing $\rho$ assumption. Notably, Si et al. (Si & Yun, 2023) establish that with a constant $\rho$, SAM cannot converge to a stationary point of the base loss $l(\theta)$, but instead oscillates within an $\mathcal{O}(\rho)$ neighborhood . However, these results are framed primarily through a static error analysis on the base loss $l(\theta)$ and treat the oscillation essentially as an irreducible optimization failure. By focusing solely on the distance to the minimum, standard analyses miss the dynamical stability inherent to the minimax objective $L(\theta)$ and the constructive role of the coupled $(\theta, \Delta)$ dynamics. In this work, we shift the analytical lens from static convergence bounds to coupled dynamical stability.

Inspired by successes of dynamical systems perspective on optimization (Su et al., 2014; Wibisono et al., 2016; Sanz Serna & Zygalakis, 2021), we derive a coupled two-timescale ODE that models the joint evolution of the learner $\theta$ and the adversary $\Delta$.

### 1.2. Contributions

1. **Lyapunov-based ODE model.** We show that the KKT conditions of the SAM bilevel game arise as equilibria of a two-timescale ordinary differential equation. A novel Lyapunov function certifies global descent of this ODE under a simple gradient-Hessian ratio condition and without assuming convexity.

2. **Uniform Error Bound for MSAM** By discretizing the ODE with separate step sizes, we obtain MSAM: a variant that performs *several* inner maximization steps before each outer update. We establish a uniform-in-time error bound between the discrete iterates and the continuous trajectory under standard Lipschitz and bounded-curvature assumptions, recovering classical SAM in the small-$\rho$ limit.

3. **Bridging intuition and practice.** Our analysis quantifies, for the first time, the gap between the ideal minimax formulation and the algorithmic SAM update as a function of $\rho$ and the loss curvature, thereby explaining why SAM and its variant succeeds in large perturbation and non-convex regimes.

4. **Empirical validation.** Experiments on synthetic functions, matrix sensing, and CIFAR-10 confirm the pre-

dicted benefits of multiple inner steps and illustrate how curvature and radius jointly control convergence and generalization.

## 2. Related Works

**Sharpness and Generalization.** Large-batch training was first shown to converge to *sharp* minima that hurt test accuracy (Keskar et al., 2017), sparking interest in the link between landscape geometry and generalization. Classical Hessian sharpness is re-parameterization-sensitive; scale-invariant alternatives correlate inconsistently with accuracy and can even anti-correlate with Out-of-Distribution (OOD) error (Andriushchenko et al., 2023). Normalization layers mitigate sharpness by driving gradient descent to an edge-of-stability regime (Lyu et al., 2022). SAM casts training as a radius-$\rho$ min-max problem and delivers strong empirical gains (Foret et al., 2021); theory has begun to explain its implicit bias in linear and nonlinear networks (Andriushchenko & Flammarion, 2022; Chen et al., 2023). Extensions such as WSAM add explicit sharpness penalties with PAC-Bayes guarantees (Yue et al., 2023), and large-scale studies confirm SAM's competitiveness in zero-shot and domain-adaptation settings (Schapiro & Zhao, 2024).

**SAM Variants.** Beyond the original algorithm (Foret et al., 2021), recent work refines curvature regularization. To achieve scale and re-parameterization invariance, ASAM utilizes adaptive frameworks and Riemannian metrics (Kwon et al., 2021), while Fisher SAM applies information geometry (Kim et al., 2022). Functional-SAM regularizes curvature in function space for better NLP performance (Singh et al., 2025). A Unified-SAM template yields convergence under Polyak-Łojasiewicz objectives (Oikonomou & Loizou, 2025), addressing theoretical gaps alongside work that rethinks SAM's fundamental optimization formulation (Xie et al., 2024). Efficiency is optimized by K-SAM and Surrogate Gap Minimization, which reduce computational overhead and refine the sharpness objective, respectively (Ni et al., 2022; Zhuang et al., 2022). SAM alone improves adversarial robustness (Zhang et al., 2024); Eigen-SAM further aligns perturbations with the top Hessian eigenvector (Luo et al., 2024). Sharpness-Aware Gradient Matching adds a third term to close the gap between empirical and perturbed losses, achieving SOTA domain-generalization (Wang et al., 2023).

**Bilevel Optimization.** Progress in bilevel and minimax optimization spans nonconvex, stochastic, and geometric settings. Two-timescale GDA converges in nonconvex-concave games and performs well in GANs (Lin et al., 2025). For nonconvex-strongly-convex bilevel problems, AID and ITD now enjoy finite-time guarantees (Ji et al., 2021a). Fully first-order stochastic methods match or improve prior sample complexities (Kwon et al., 2023; Chen et al., 2022),

while perturbed AID escapes saddle points with high probability (Huang et al., 2025). Optimality conditions have been extended to constrained and Riemannian settings (Dai & Zhang, 2020; Li & Ma, 2025), and control-theoretic "safe gradient flows" enforce feasibility in a single loop (Sharifi et al., 2025).

**ODE and Lyapunov Analysis.** Viewing optimization algorithms as time-discretizations of ODEs has shed light on many accelerated methods. The vanishing-step limit of Nesterov's scheme yields a second-order ODE (Su et al., 2014); Bregman Lagrangians and high-resolution flows explain broader families and discretization artifacts (Wibisono et al., 2016; Shi et al., 2022). Estimate sequences are equivalent to Lyapunov functions in discrete and continuous time (Wilson et al., 2021a;b), and LMI tools provide systematic certificates (Sanz Serna & Zygalakis, 2021; Dobson et al., 2025). Weak discrete gradients unify discretizations (Lu, 2021; Ushiyama et al., 2023); stochastic extensions recover accelerated rates under noise (Laborde & Oberman, 2020; Moucer et al., 2023).This continuous-time perspective has recently been extended to define a universal class of SAM algorithms, providing a unified framework for their convergence and stability (Tahmasebi et al., 2024). Notably, trajectory analysis reveals that these algorithms do not merely minimize sharpness, but achieve better generalization through specific implicit biases inherent in their dynamics (Wen et al., 2023). Recent work designs safe gradient flows via control barrier functions (Allibhoy & Cortés, 2024) and shows that natural data variation can remove spurious local trajectories (Fattahi et al., 2023). Lyapunov learning has also improved Neural-ODE training and safety certification (Rodriguez et al., 2022; Richards et al., 2018), while differential-equation viewpoints continue to generate new accelerated algorithms (Siegel, 2021).

## 3. Theoretical Analysis

### 3.1. Notations and Assumptions

We use $\|\cdot\|_2$ to denote the $\ell_2$ norm for vectors and Frobenius norm for matrices, and $\|\cdot\|_{op}$ to denote the operator norm for matrices. $I$ denotes the identity matrix. $\sigma_i(\cdot)$ denotes the $i$-th largest singular value of a matrix. $\lambda_i(\cdot)$ denotes the $i$-th largest eigenvalue of a symmetric matrix. For a twice-differentiable $f : \mathbb{R}^d \to \mathbb{R}$, let $\nabla f(\cdot)$ and $\nabla^2 f(\cdot)$ denote its gradient and Hessian, respectively.

**Assumption 3.1.** The loss function $l : \mathbb{R}^d \to \mathbb{R}$ is twice continuously differentiable.

### 3.2. KKT Conditions and Idealized Dynamics

In this section, we first derive the KKT conditions of the inner maximization from defined in (1b), and derive the idealized dynamics that would result from perfectly solving the min-max game (1).

We begin by characterizing the geometry of the optimal perturbation. While the inner maximization is generally non-concave, the continuity assumption on $l$ over the compact feasibility set $\{\Delta \colon \|\Delta\|_2 \leq \rho\}$ guarantees the existence of a global maximum. Furthermore, any local maximizer must satisfy the following first-order necessary conditions for $\Delta$ to be an optimal solution for problem (1b) given in the following lemma.

**Lemma 3.2.** *If $\Delta$ is a local maximizer of problem (1b), then $\Delta$ must satisfy either of the following conditions:*

$$\Delta = \rho \, \frac{\nabla l(\theta + \Delta)}{\|\nabla l(\theta + \Delta)\|_2} \tag{2a}$$

$$or \qquad \nabla l(\theta + \Delta) = 0 \tag{2b}$$

See Appendix A for the proof.

In the case where $\Delta$ is a local maximizer and condition (2b) holds, $\theta$ is forced to be a local maximizer of the problem (1a), which contradicts our minimization setting over $\theta$. Hence, we will focus on the former case of condition (2a), which recovers the one-step update rule of the normalized SAM algorithm.

Let $\Delta^*(\theta)$ be an optimal solution implicitly defined by (2a). By applying the chain rule, the time evolution of $\theta$ is governed by the total derivative:

$$
\begin{aligned}
\frac{d\theta}{dt} &= -\nabla_\theta L(\theta) \\
&= -\left(I + \nabla_\theta \Delta^*(\theta)^\top\right) \nabla l(\theta + \Delta^*(\theta))
\end{aligned}
\tag{3}
$$

where $\nabla_\theta \Delta^*(\theta)$ is the Jacobian of the optimal perturbation with respect to $\theta$. While Eq. (3) explicitly includes the Jacobian term $\nabla_\theta \Delta^*(\theta)^\top \nabla l^*$, this term vanishes under the optimality condition of the inner maximization: $\nabla_\theta \Delta^{*\top} \nabla l^* = 0$. This result, known as Danskin's Theorem, implies that the gradient of the value function simplifies to $\nabla_\theta L(\theta) = \nabla l(\theta + \Delta^*(\theta))$. Thus, the dynamic is:

$$\frac{d\theta}{dt} = -\nabla l(\theta + \Delta^*(\theta)) \tag{4}$$

### 3.3. Two-timescale ODE

While Eq. (4) provides a useful baseline, it fails to capture the practical realities of the optimization landscape for two reasons. Firstly, in non-convex deep learning landscapes, the optimal perturbation $\Delta^*(\theta)$ is not guaranteed to be unique or continuous. As the optimizer traverses regions with multiple local maxima for the adversary, $\Delta^*(\theta)$ may jump discontinuously, rendering the vector field in Eq. (4) non-differentiable and the dynamics ill-posed. Secondly, practical implementations like SAM do not access the oracle $\Delta^*(\theta)$ instantaneously.

Viewing the SAM inner update as the fixed point iteration of solving the first-order necessary condition (2a) motivates us to lift the analysis to the joint space $(\theta, \Delta)$. Let $\theta(t)$ and $\Delta(t)$ represent the parameter and the perturbation as a function of continuous time $t$. We use the coupled ODE to model a continuous pursuit game: the parameter $\theta(t)$ descends the loss landscape under current perturbation $\Delta(t)$, while $\Delta(t)$ actively chases the moving target of the optimal perturbation for the current $\theta(t)$.

Below we formalize our derivation: let $\Delta(t)$ represent the perturbation as a function of the continuous time $t$. Given the fixed-point iteration $\Delta_{t+1} = \rho \frac{\nabla l(\theta + \Delta_t)}{\|\nabla l(\theta + \Delta_t)\|_2}$, we introduce a small time step $h$ to derive an ODE approximation:

$$\Delta(t+h) - \Delta(t) = h \left( \rho \cdot \frac{\nabla l(\theta + \Delta(t))}{\|\nabla l(\theta + \Delta(t))\|_2} - \Delta(t) \right) \quad (5)$$

As $h \to 0$, the difference $(\Delta(t + h) - \Delta(t))/h$ can be approximated by the derivative $\frac{d\Delta}{dt}$. Thus, the continuous flow for the fixed-point iteration can be written as:

$$\frac{d\Delta}{dt} = \rho \cdot \frac{\nabla l(\theta + \Delta)}{\|\nabla l(\theta + \Delta)\|_2} - \Delta \quad (6)$$

At the same time, we have an additional norm constraint from (14d) on $\Delta(t)$, requiring $\Delta(t)$ to have a fixed norm of $\rho$, which is

$$\frac{d\|\Delta(t)\|_2^2}{dt} = 0 \quad \Rightarrow \quad \Delta(t)^\top \frac{d}{dt}(\Delta(t)) = 0 \quad (7)$$

Combining the iterative step in (2a) and the fixed norm constraint from (14d), we make an additional step to project $\frac{d\Delta}{dt}$ in (6) to the tangent space of the sphere $\|\Delta\|_2 = \rho$. By defining $P_\Delta^\perp = I - \frac{\Delta\Delta^\top}{\|\Delta\|_2^2}$, the flow of the system in continuous time can be described by the following ODE:

$$\begin{aligned}
\frac{d\Delta}{dt} &= P_\Delta^\perp \left( \rho \cdot \frac{\nabla l(\theta + \Delta)}{\|\nabla l(\theta + \Delta)\|_2} - \Delta \right) \\
&= \rho \cdot \frac{\nabla l(\theta + \Delta)}{\|\nabla l(\theta + \Delta)\|_2} - \frac{\rho \Delta\Delta^\top \nabla l(\theta + \Delta)}{\|\Delta\|_2^2 \|\nabla l(\theta + \Delta)\|_2}
\end{aligned} \quad (8)$$

We now formulate the complete dynamics by combining the projection-based ODE for $\Delta$ (derived in Eq. 8) with the parameter dynamics. We introduce timescale parameters $\alpha, \beta$ to control that the perturbation $\Delta$ evolves faster than $\theta$:

$$\begin{cases}
\frac{d\theta}{dt} = -\alpha \nabla l(\theta + \Delta) \\
\frac{d\Delta}{dt} = \beta P_\Delta^\perp \left( \rho \cdot \frac{\nabla l(\theta + \Delta)}{\|\nabla l(\theta + \Delta)\|_2} - \Delta \right)
\end{cases} \quad (9)$$

The ODE given in (9) describes the continuous-time evolution of $\Delta$ toward the fixed points defined by the iterative (2a), while $\theta(t)$ descends the loss surface evaluated at this worst-case point. These are also equilibrium points of the ODE

(where $\frac{d\Delta(t)}{dt} = \frac{d\theta(t)}{dt} = 0$). As practical implementations of SAM rely on a finite number of inner ascent steps $M$, which in the continuous-time limit, translates to a finite adversarial response speed $\beta$, the coupled two-timescale ODE (9) captures that the adversary is lagging behind. Also, asymptotically, as $\beta/\alpha \to \infty$, (9) recovers the idealized ODE (4). Below we provide the convergence theorem for the two-timescale ODE (9).

**Theorem 3.3.** *Consider the $\theta$- and $\Delta$-dynamics as defined in (9) initialized with any feasible perturbation $\|\Delta_0\|_2 \leq \rho$. The trajectory $(\theta_t, \Delta_t)$ asymptotically converges to the set $\Omega \subset \mathbb{R}^{d \times d}$ defined by:*

$$\Omega = \{(\theta, \Delta) : \|\nabla l(\theta + \Delta)\|_2 \leq \rho \cdot \max\{C, \sigma_1(\nabla^2 l(\theta + \Delta))\}\}$$

*where $C > 0$ is a constant dominated by $\beta/\alpha$.*

The theorem is proved based on a Lyapunov analysis. See Appendix C for more details. Notice that the convergence set $\Omega$ is defined jointly by the perturbation drift and the local curvature scale. In regions of low curvature $\sigma_1(\nabla^2 l(\theta + \Delta)) \ll \frac{\beta}{\alpha}$, the convergence is limited by the linear drift $\frac{\beta}{\alpha}\rho$. While in regions of high curvature that $\sigma_1(\nabla^2 l(\theta + \Delta)) > \frac{\beta}{\alpha}$, if the high curvature is local as that around a sharp local minimum, the violation of the stability condition $\|\nabla l(\theta + \Delta)\| \leq \rho \sigma_1(\nabla^2 l(\theta + \Delta))$ manifests as transient instability forcing the trajectory out of high-curvature regions. The adversarial pressure increases the Lyapunov energy, effectively bounce the trajectory out of the sharp region until it finds a basin flat enough to sustain stable convergence.

The $\beta/\alpha$ ratio governs a trade-off between adversarial tracking fidelity and convergence precision. A large $\beta/\alpha$ ensures $\Delta(t) \approx \Delta^*(\theta(t))$, providing an accurate estimate of the local sharpness. However, its asymptotic convergence region contains an $O(\rho)$ term growing with $\beta/\alpha$ from the residual threshold in the Lyapunov characterization. This reveals a trade-off: larger $\beta/\alpha$ improves inner-tracking fidelity but reduces convergence precision. Thus, increasing $\beta/\alpha$ is not guaranteed to improve final performance, despite improving approximation to the idealized ODE. On complex tasks like CIFAR-100, where filtering sharp minima outweighs the loss of optimization precision, running more iterations of the inner loop can improve test accuracy (Foret et al., 2021). For our synthetic experiments on spurious matrix completion in Section 4.1.2, we observe that larger $M$ does not necessarily lead to better convergence.

Since the convergence set $\Omega$ has a non-zero measure scaling with $\rho$, Theorem 3.3 implies that for a fixed perturbation radius $\rho > 0$, the system cannot asymptotically converge to a singleton critical point (where $\|\nabla l\| = 0$). Instead, the coupled $(\theta, \Delta)$ dynamics are forced into a state of bounded oscillation or a limit cycle within $\Omega$. This suggests that

"convergence" in SAM is effectively a dynamic equilibrium where the trajectory orbits the minimum, driven by the persistent repulsive force of the adversarial perturbation. To achieve true stationarity, an annealing schedule where $\rho \to 0$ is theoretically required.

### 3.4. Discrete Algorithm Error Bound

To theoretically quantify the deviation between our continuous-time formulation defined in Eq. (3) and practical implementations, we must analyze the discretization error. Specifically, we aim to understand how the trajectory of the standard SAM algorithm diverges from the ideal minimax solution defined by the ODE.

To facilitate a rigorous comparison, we define the MSAM. This formulation allows us to vary the timescale separation explicitly via the number of inner steps $M$, with Standard SAM emerging as the special case where $M = 1$.

---

**Algorithm 1** Multiple-Step SAM (MSAM)

---

1: **Require:** learning rates $\eta$, radius $\rho$, inner iterations $M$
2: **Repeat until convergence:**
3: **for** $m = 1$ **to** $M$ **do**
4:     Update $\Delta$:    $\Delta \leftarrow \rho \nabla l(\theta + \Delta)/\|\nabla l(\theta + \Delta)\|_2$
5: **end for**
6: Update $\theta$:    $\theta \leftarrow \theta - \eta \nabla l(\theta + \Delta)$

---

With this definition, we can now state the main error bound relating the discrete MSAM trajectory to the continuous ODE. Below are the standard smoothness conditions required by our convergence analysis:

**Assumption 3.4.** The gradient of $l$, $\nabla l(\cdot)$ and hessian of $l$, $\nabla^2 l(\cdot)$ are Lipschitz continuous, i.e.,

$$\|\nabla l(w) - \nabla l(w')\| \leq L\|w - w'\|,$$
$$\|\nabla^2 l(w) - \nabla^2 l(w')\| \leq L\|w - w'\|$$

**Assumption 3.5.** The k-th order derivative of $l$, $\nabla^k l$, is norm bounded: $\|\nabla^k l(w)\| \leq B_k$ for $k = 1, 2$ and $w$ in the region R.

**Theorem 3.6** (Error Bound for Finite-Step MSAM). *Let $\theta(t)$ be the solution to the ideal Reduced ODE $\frac{d\theta}{dt} = -\nabla l(\theta + \Delta^*(\theta))$. Let $\{\theta_k\}$ be the sequence generated by MSAM (Algorithm 1) with a fixed number of inner iterations $M \geq 1$. Assume the inner fixed-point map $T_\theta(\Delta)$ is a contraction with rate $\kappa < 1$ (i.e., local curvature condition). Under Assumption 3.4 and 3.5, for a fixed time horizon $T$ (with $N\eta = T$), the global error satisfies:*

$$\sup_{0 \leq k \leq N} \|\theta_k - \theta(t_k)\| \leq C_1 \cdot \eta + C_2 \cdot \kappa^M$$

*where $C_1$ and $C_2$ are constants depending on the smoothness properties of the loss and the time horizon $T$.*

See Appendix D for the proof. Let $c = \inf_{w \in R} \|\nabla l(w)\|$ be the minimum gradient norm, $\kappa \triangleq \frac{\rho B_2}{c}$ is the contraction ratio of the inner fixed-point map. When $M = 1, \rho < 1, \kappa < 1$, Algorithm 1 reduces to the standard SAM, which matches the $\mathcal{O}(1)$ convergence error under PL condition.

The above theorem validates that SAM's empirical success stems from approximating min-max stationary points while neglecting higher-order terms. These bounds reveal critical insights into the stability of MSAM:

- **The Stability Threshold ($\kappa < 1$):** The analysis holds only when $\rho B_2 < c$. This implies that the perturbation radius $\rho$ must be small relative to the "curvature-to-gradient" ratio. Physically, the gradient must provide a strong enough restoring force to overcome the destabilizing effect of the Hessian.

- **Singularity at High Curvature:** As $\kappa \to 1$ (e.g., near sharp saddle points or if $\rho$ is too large), the term $1 - \kappa$ approaches zero, causing error bounds to diverge. This causes the error constants $C_1, C_2$ to diverge, reflecting the fact that the worst-case direction becomes ill-defined or highly sensitive when the local curvature dominates the gradient signal.

- **Practical Trade-off:** There is a tension between the radius $\rho$ and the error bound. While a larger $\rho$ is desired for better generalization (sharper minima avoidance), it linearly increases the condition number $\frac{1}{1-\kappa}$, thereby requiring a finer step size $\eta$ and more inner steps $M$ to maintain the same approximation accuracy. $M$ must increase logarithmically for constant fixed-point error. As discretization error scales with $1 - \kappa$, $\eta$ must decrease proportionally.

To maintain a fixed approximation error level $\varepsilon$ (i.e., ensuring $C_2 \kappa^M \approx \varepsilon$), the number of required iterations scales as $M \approx \frac{\ln(\varepsilon/C_2)}{\ln \kappa}$. In terms of wall-clock time, MSAM's cost scales linearly with the inner loop depth $M$, making the choice of $M$ (and by extension $\rho$) a critical regulator between computational efficiency and the fidelity of the curvature approximation.

### 3.5. Connection to Penalized Gradient Norm

Despite the simplification provided by Danskin's Theorem, explicitly calculating $\nabla l(\theta + \Delta^*)$ requires solving the inner loop. To interpret the dynamics without solving the inner loop, we analyze the first-order Taylor expansion of this gradient term around $\theta$. This reveals a structural equivalence to Approximate Implicit Differentiation (AID) methods used

in bilevel optimization:

$$\nabla l(\theta + \Delta^*) \approx \nabla l(\theta) + \nabla^2 l(\theta)\Delta^*$$

$$\approx \nabla l(\theta) + \nabla^2 l(\theta)\rho \frac{\nabla l(\theta)}{\|\nabla l(\theta)\|_2}$$

$$= \nabla l(\theta) + \underbrace{\nabla^2 l(\theta)}_{\substack{\text{Cross-Deriv.}\\ \nabla^2_{xy}}} \underbrace{\left(\frac{\rho}{\|\nabla l(\theta)\|_2}I\right)}_{\substack{\text{Effective}\\ \mathcal{H}^{-1}}} \underbrace{\nabla l(\theta)}_{\substack{\text{Inner Grad.}\\ \nabla_y}}$$

$$(10)$$

In general bilevel problems $\min_x F(x, y^*(x))$ as in (Ji et al., 2021b), the update direction is $\nabla_x F - \nabla^2_{xy}F[\nabla^2_{yy}F]^{-1}\nabla_y F$. As shown in Eq. (10), our expansion recovers this form exactly. The scalar matrix $\frac{\rho}{\|\nabla l\|_2}I$ acts as the effective Inverse Hessian $[\nabla^2_{yy}F]^{-1}$ for the constrained problem. This term arises from the KKT conditions of the spherical constraint: the Lagrange multiplier scales as $\lambda \approx \|\nabla l\|_2/\rho$, creating an isotropic stiffness that resists the gradient force, resulting in an effective inverse curvature of $(\lambda I)^{-1}$. A detailed discussion can be found in Appendix B.

This interpretation suggests that SAM dynamics effectively operate as if the update is preconditioned by $(\lambda I - \nabla^2 l)^{-1}$ in a proximal-like sense, yet it differs fundamentally from standard preconditioning methods like Newton's method or RMSProp. Instead of accelerating along low-curvature directions, this effective preconditioning suppresses updates where the gradient is highly sensitive to perturbations (high $\nabla^2$). Consequently, the trajectory is biased toward regions where the landscape is locally flat, prioritizing stability over the pure descent speed favored by second-order optimizers.

The explicit gradient derived in Eq. (10) not only unifies bilevel optimization concepts but also provides a rigorous theoretical foundation for heuristically proposed regularization methods. Specifically, our derived update direction is mathematically equivalent to the gradient of the **Penalized Gradient Norm (PGN)** objective proposed by (Zhao et al., 2022):

$$\mathcal{L}_{PGN}(\theta) = l(\theta) + \rho\|\nabla l(\theta)\|_2$$

$$\implies \quad \nabla\mathcal{L}_{PGN} = \nabla l(\theta) + \rho\frac{\nabla^2 l(\theta)\nabla l(\theta)}{\|\nabla l(\theta)\|_2} \quad (11)$$

In practice, computing the explicit Hessian-vector product is computationally expensive. Therefore, we approximate the curvature term using a finite difference quotient. Substituting this approximation into the update direction yields the practical update rule:

$$\nabla_\theta L(\theta) = (1-\alpha)\nabla_\theta L(\theta) + \alpha\nabla_\theta L\left(\theta + r\frac{\nabla_\theta L(\theta)}{\|\nabla_\theta L(\theta)\|}\right) \quad (12)$$

This formulation reveals that the effective gradient is simply a weighted average of the current gradient and the lookahead

gradient at the perturbed point, avoiding explicit second-order computations while retaining curvature information.

*Remark* 3.7. Regarding division by $\|\nabla l(\theta)\|_2 = 0$, the singularity is inherent to normalized SAM. As the perturbation direction is normalized by a gradient norm, it can arise near stationary points. In our notation, $w = \theta + \Delta$. Outside the convergence region $\Omega$, we have by definition $\|\nabla\ell(w)\| > \max\{C\rho, \rho\sigma_1(\nabla^2\ell(w))\} \geq C\rho$, so the shifted gradient norm is uniformly bounded away from zero. In practice, standard implementations use $\epsilon$-smoothing, replacing $\|\nabla\ell(w)\|$ by $\|\nabla\ell(w)\| + \epsilon$ to avoid numerical issue.

# 4. Empirical Evaluation

## 4.1. Synthetic Data

### 4.1.1. VISUALIZATION OF OPTIMIZATION TRAJECTORIES ON A NON-CONVEX LANDSCAPE

We consider a synthetic two-dimensional loss function with multiple minima, defined as

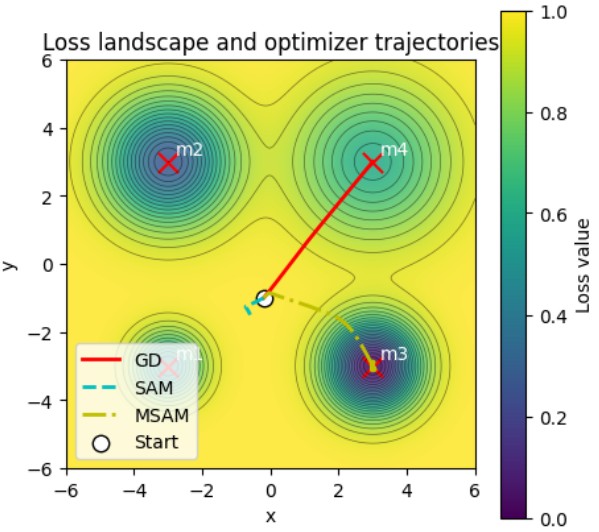

*Figure 1.* Loss landscape and optimizer trajectories

$$L(x, y) = \sum_{i=1}^{n} -A_i \exp\left\{-\frac{(x - c_{i,x})^2 + (y - c_{i,y})^2}{2\sigma_i^2}\right\}, \quad (13)$$

where $(c_{i,x}, c_{i,y})$ are the centers of the minima, $A_i$ are their amplitudes (depths), and $\sigma_i$ are the curvatures. This function creates a landscape with minima of different depth and sharpness, allowing us to study the convergence behavior of various optimization algorithms.

We evaluate three optimization algorithms: standard gradient descent (GD), Sharpness-Aware Minimization (SAM), and multi-step SAM (MSAM). All algorithms are initialized from the same starting point $\mathbf{x}_0 = (-0.2, -1)$ and use a

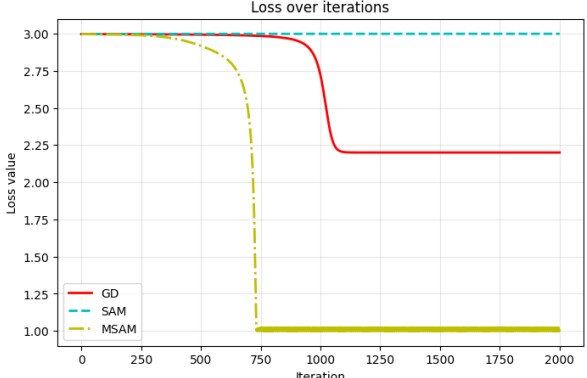

*Figure 2.* Loss over iterations

fixed learning rate of 0.1. Both SAM and MSAM use a perturbation magnitude $\rho = 1$, while MSAM performs 10 inner steps per iteration to better approximate the adversarial gradient.

Figure 1 shows the loss landscape and the optimizer trajectories. We observe that: GD quickly converges to a local minimum near the starting point, failing to reach the global minimum. SAM oscillates and does not converge to any minimum due to large sharpness-aware perturbations. While at the same time, MSAM successfully navigates the landscape and converges to the global minimum, benefiting from the multi-step adversarial gradient to escape local minima.

Figure 2 plots the loss values over iterations, which quantitatively confirms the convergence behaviors observed in the trajectories. MSAM achieves the global minima, while GD is trapped in a suboptimal local minimum and SAM exhibits oscillatory behavior. As discussed in (Kim et al., 2023), SAM fails to escape saddle in non-convex landscapes with large $\rho$, highlighting the misalignment of minimax goal and SAM update. Instead, by carefully choosing $M$ and $\eta$, large $\rho$ becomes a practical choice.

Moreover, we re-run the experiment with $\rho = 0.6$ and increase $M$. The results reveal a clear phase transition: when $M = 1$, the trajectory sticks at a saddle point. When $M = 2, 3$, it escapes the saddle but converges to a suboptimal local minimum (like GD). When $M \geq 4$, it successfully navigates the complex landscape to the true global minimum. We exhibit such a regime where insufficient inner accuracy leads to wrong local approximation, and where increasing $M$ beyond a threshold recovers the intended minimax dynamics and reaches the optimal solution, validating Theorem 3.6. Note that larger $M$ is not universally better. Additional discussions are provided in Appendix F.5.

### 4.1.2. SPURIOUS MATRIX COMPLETION PROBLEM

We also use matrix completion problems to test the convergence behaviors of SAM and MSAM in large-$\rho$ regime

and the performance of SAM and MSAM in escaping local minima in a spurious optimization landscape. Consider the matrix completion problem

$$\min_{\mathbf{x} \in \mathbb{R}^n} f(\mathbf{x}; \epsilon) := \|(\mathbf{x}\mathbf{x}^\top - \mathbf{M}(\epsilon)^*)_\Omega\|_F,$$

where $\mathbf{M}^* \in \mathbb{R}^{n \times n}$ is a rank-1 matrix generated by $\mathbf{M}^* = \mathbf{x}^* \mathbf{x}^{*\top}$ for $\mathbf{x}^* \in \mathbb{R}^n$. Then $\mathbf{M}(\epsilon)^* = (\mathbf{x}^* + \epsilon)(\mathbf{x}^* + \epsilon)^\top$, where $\epsilon \in \mathbb{R}^n$ is a small perturbation. Additionally, $\Omega$ denotes the set of indices of observed entries, and $F$ denotes the Frobenius norm of a matrix. We generate $\mathbf{x}^*$ in a similar way as Examples 1 and 2 given in (Yalçın et al., 2022). Then these matrix completion problems are particularly hard because of the existence of exponentially many local minima.

**Example 1** We take $\mathbf{x}^* \in \mathbb{R}^n$ such that $\mathbf{x}_i^* = 1$ for even values of $i$, and $\mathbf{x}_i^* = 0$ for odd values of $i$. The set of observed entries is $\Omega = \{(i, j) : |i - j| \leq 1, i, j \in [n]\}$. We solve the matrix optimization problem with the standard gradient descent, SAM, and MSAM for $M \in \{5, 10\}$. For SAM and MSAM, we take $\rho \in \{0.1, 0.2, 0.3, 0.35, 0.4\}$. All experiments are run for $n \in \{10, 100, 1000\}$, learning rate 0.01, and 2000 iterations. Each entry in the small perturbation $\epsilon$ is sampled from the standard normal distribution and then rescaled to the desired noise level. The noise level is set to 0.005. The optimized loss results are summarized in Tables 1, 2, 3, 4.

*Table 1.* Loss for GD, SAM $n = 100$

| $\rho$ | GD | SAM |
|---|---|---|
| 0.20 | 6.9556e-06 | 2.9435e-05 |
| 0.30 | 6.9556e-06 | 0.0006 |
| 0.40 | 1.0479e-05 | 1.5027 |

*Table 2.* Loss for GD, MSAM, $n = 100$

| $\rho$ | GD | MSAM (M=5) | MSAM (M=10) |
|---|---|---|---|
| 0.20 | 6.9556e-06 | 2.2058e-05 | 1.5192e-05 |
| 0.30 | 6.9556e-06 | 1.3513e-05 | 0.00025 |
| 0.40 | 1.0479e-05 | 7.5121e-05 | 0.00056 |

*Table 3.* Loss for GD, SAM, $n = 1000$

| $\rho$ | GD | SAM |
|---|---|---|
| 0.30 | 1.2474e-05 | 1.2932e-05 |
| 0.35 | 1.2474e-05 | 0.0006 |
| 0.40 | 1.1989e-05 | 0.0006 |

*Table 4.* Loss for GD, MSAM, $n = 1000$

| $\rho$ | GD | MSAM (M=5) | MSAM (M=10) |
|---|---|---|---|
| 0.30 | 1.2474e-05 | 8.9293e-05 | 8.8327e-05 |
| 0.35 | 1.2474e-05 | 0.0001 | 6.9054e-05 |
| 0.40 | 1.1989e-05 | 0.0001 | 9.0831e-05 |

*Table 5.* Loss, Distance from $\mathbf{x}^* + \epsilon$ for GD, SAM, $n = 10$, $\epsilon = 0.5$

| $\rho$ | GD | SAM |
|------|------|------|
| 0.20 | 0.2485, 3.4448 | 2.5127, 2.1884 |
| 0.40 | 0.2485, 3.4448 | 2.5120, 2.1879 |
| 0.50 | 0.2485, 3.4448 | 2.5112, 2.1937 |

*Table 6.* Loss, Distance from $\mathbf{x}^* + \epsilon$ for MSAM, $n = 10$, $\epsilon = 0.5$

| $\rho$ | MSAM (M=5) | MSAM (M=10) |
|------|------|------|
| 0.20 | 2.2726, 2.0294 | 0.2487, 3.4613 |
| 0.40 | 2.5128, 2.1816 | 0.0549, 2.1761 |
| 0.50 | 2.5129, 2.1937 | 0.0576, 2.1716 |

These results show that under an appropriate choice of $M$, MSAM converges to at least the same order as GD and SAM. Also, notice that in the large $\rho$ regime, when SAM does not converge to any minimum, MSAM still converges to at least a local minimum and matches up with the performance of GD. When the problem scale goes from 100 to 1000, and $\rho$ from 0.3 to 0.4, we observe that larger $M$ is needed for the algorithm to even converge. This validates theorem 3.6 that we need more iterations, i.e. larger $M$, to reduce the fixed point iteration error $\kappa^M$, where $\kappa$ scales linearly with $\rho$. Another remarkable phenomenon is that for the moderate scale problem $n = 100$, running MSAM with $M = 5$ outperforms that with $M = 10$, this validates the effectiveness of the stability condition $\|\nabla l(\theta + \Delta)\|_2 \geq (\beta/\alpha)\rho$ in theorem 3.3. Larger $M$ corresponds to larger $\beta/\alpha$, implying that the adversary catches up the motion of parameters faster, making the exact convergence to the optimal parameters harder.

**Example 2** We take $\mathbf{x}^* \in \mathbb{R}^n$ such that $\mathbf{x}_i^* = 1$ for $i$ odd, and $\mathbf{x}_i^* = 0$ for $i$ even. The set of observed entries is $\Omega = \{(i, i), (i, 2k), (2k, i) : \forall i \in [n], k \in [\lfloor n/2 \rfloor]\}$. The results of (Yalçın et al., 2022) indicate that $\mathbf{x}^* + \epsilon$ is a unique global minimum up to a sign flip for the perturbed problem. Therefore, in addition to the optimized loss, we also report $\|\mathbf{x} - \mathbf{x}^*\|_2$, where $\mathbf{x}$ is the output vector.

We solve the matrix optimization problem with the standard gradient descent, SAM, and MSAM for $m \in \{5, 10\}$. All experiments are run for $n = 10$, learning rate 0.01, and 2000 iterations. Each entry in the small perturbation $\epsilon$ is sampled from the standard normal distribution and then rescaled to the desired noise level. The noise level is set to 0.5 respectively. The results are summarized in Tables 5 and 6. The results show that when $\rho$ is large, MSAM outperforms both GD and SAM in the sense that besides substantially reducing the loss function, MSAM also converges closer to the global minima.

### 4.2. Experiments on CIFAR-10 and CIFAR-100

We would like to refer to Appendix in (Foret et al., 2021), where statistically significant improvement of MSAM of over is provided. Here we would like to elaborate more on its effect for different $\rho$, and M. Additional experiments are provided in Appendix F.6.

**Experimental Setup:** We evaluate Multi-step Sharpness-Aware Minimization (MSAM) on CIFAR dataset (Krizhevsky, 2009) using ResNet20, ResNet56, and Wide ResNet28-2 architectures. All models are trained for 200 epochs with cosine annealing learning rate (initial LR=0.1), batch size 128, momentum=0.9, and weight decay=5e-4. We compare Stochastic Gradient Descent (SGD) baseline, standard SAM ($M = 1$), and MSAM variants ($M \in \{3, 5, 10\}$) across neighborhood sizes $\rho \in \{0.025, 0.05, 0.10\}$. Deeper architectures (ResNet56 and WRN28-2) are evaluated with $\rho \in \{0.05, 0.10\}$ to assess generalization across network depths.

**Results and Discussion:** Tables 7, 8, 9 present test accuracy across configurations. One can see that SAM and MSAM consistently outperform SGD across all architectures, with accuracy gains of +0.2–0.6% for ResNet20 and +0.2–0.4% for deeper networks. Moreover, in small $\rho$ settings, MSAM achieves comparable performance to SAM, and in moderate $\rho$ setting, there exists larger improvement, validating the multi-step approximation's effectiveness. Notably, under CIFAR-100 like MSAM (M=5) even surpasses SAM on ResNet56 at $\rho = 0.10$ (74.8% vs 74.2%).

**Theoretical Interpretation:** These results strongly support our min-max framework. The statistically significant superiority of SAM/MSAM over SGD aligns with their convergence to min-max stationary points of $\min_\theta \max_{\|\Delta\| \leq \rho} L(\theta + \Delta)$. MSAM's performance parity with SAM confirms our theoretical insight that multiple fixed-point iteration steps preserve the stationary point approximation. The problem specific $\rho$ and $M$ sensitivity reflect our theoretical prediction: complex tasks (CIFAR-100) benefit more from MSAM where strong non-convexity dominates, while easier tasks (CIFAR-10) operate in flatter regions where small $\rho$ and $M$ suffice. This validates our explanation that SAM's success stems from min-max optimization rather than specific implementation choices.

*Table 7.* Test Accuracy on CIFAR-10 Comparing MSAM and SGD

| ResNet20 | SGD | M=3 | M=5 | M=10 |
|------|------|------|------|------|
| $\rho = 0.025$ | 93.2 | 93.4 | 93.6 | 93.3 |
| $\rho = 0.05$ | 93.2 | 93.3 | 93.4 | 93.4 |
| $\rho = 0.10$ | 93.2 | 93.7 | 93.7 | 93.6 |

*Table 8.* Test Accuracy on CIFAR-10 of Wide ResNet28 and ResNet56

| WRN28-2 | SGD | Standard SAM | M=5 |
|---|---|---|---|
| $\rho = 0.05$ | 95.4 | 95.8 | 95.8 |
| $\rho = 0.10$ | 95.4 | 95.6 | 95.8 |
| ResNet56 | SGD | Standard SAM | M=5 |
| $\rho = 0.05$ | 95.0 | 95.3 | 95.2 |
| $\rho = 0.10$ | 95.0 | 95.2 | 95.4 |

*Table 9.* Test Accuracy of CIFAR-100

| CIFAR-100 | SGD | Standard SAM | M=5 |
|---|---|---|---|
| $\rho = 0.05$, ResNet20 | 70.1 | 70.7 | 70.8 |
| $\rho = 0.10$, ResNet20 | 71.2 | 71.3 | 71.4 |
| $\rho = 0.05$, ResNet56 | 75.2 | 75.2 | 75.2 |
| $\rho = 0.10$, ResNet56 | 74.4 | 74.2 | 74.8 |

*Table 10.* Performance and timing comparison of Standard SAM versus Periodic MSAM on ResNet-20.

| Method | Inner Steps ($M$) | $\rho$ (Last 20% Epochs) | Test Accuracy | Total Time |
|---|---|---|---|---|
| Standard SAM | 1 | 0.05 (All epochs) | 69.32% | 22m 5s |
| Periodic MSAM | 5 | 0.2 | 69.76% | 27m 42s |
| Periodic MSAM | 10 | 0.1 | 70.01% | 31m 41s |
| Periodic MSAM | 10 | 0.2 | 70.32% | 28m 1s |

### 4.3. Periodic MSAM

To address the computational overhead while preserving the theoretical benefits of multiple inner steps, we implemented a Periodic MSAM strategy. Recognizing that standard SAM is sufficient for the simpler, early stages of optimization, we train ResNet-20 using standard SAM ($\rho = 0.05$) for the first 80% of the training epochs. During the final 20% of the epochs (epochs 160-200), where the landscape becomes highly non-convex and the optimizer is more likely to encounter spurious local minima, we dynamically switch to MSAM with a larger perturbation radius ($\rho = 0.2$) and multiple inner steps ($M$) to accurately resolve the worst-case landscape. We also include a specific ablation using $\rho = 0.1$ for $M = 10$ during this final phase. As shown in Table 10, Periodic MSAM consistently improves test accuracy while keeping the additional wall-clock training time highly manageable.

Crucially, these results empirically validate our theoretical claims: employing a larger $\rho$ directly benefits final test accuracy by enforcing a wider flatness penalty. Furthermore, increasing $M$ provides distinct benefits when $\rho$ is large; raising $M$ from 5 to 10 under $\rho = 0.2$ further boosts accuracy from 69.76% to 70.32%, confirming that sufficient inner steps are necessary to unlock the generalization benefits of a large perturbation radius.

Regarding the scalability of this approach, we evaluated the proposed Periodic MSAM against a standard SAM baseline using a ResNet-50 model on the Imagenette2 (Howard, 2019) dataset. Both models were trained from scratch for 90 epochs using a batch size of 256, a base learning rate of 0.1 with a cosine annealing schedule, momentum of 0.9, and weight decay of 1e-4. While the baseline SAM ($\rho = 0.1$) achieved a final Top-1 validation accuracy of 75.01%, our Periodic MSAM strategy achieved 76.87%. This significant +1.86% improvement was realized by conditioning the network with standard SAM for the first 80% of training, followed by a dynamic transition to a multi-step MSAM exploration phase ($\rho = 0.2, M = 5$) for the final 20%. These results empirically validate that engaging a heavier, multi-step exploration phase in the late stages of training successfully navigates complex loss landscapes and prevents suboptimal convergence in larger-scale vision tasks.

## 5. Conclusion

We showed that SAM can be viewed as solving a minimax game and built a short, two-step mathematical model that explains how it works without assuming the loss is favorably curved or the perturbation is sufficiently small. We validate our theoretical intuition through *Multi-step SAM*, which maintains SAM's speed but has better stability for large $\rho$. Our theory pinpoints how the perturbation size and the number of inner steps control both training cost and the flatness of the final solution, and the experiments support our findings. Our analysis could help the user finetune SAM automatically, combine it with adaptive optimizers, and handle data-dependent perturbations in large-scale deep learning.

### Impact Statement

This paper presents work whose goal is to advance the field of Machine Learning. There are many potential societal consequences of our work, none which we feel must be specifically highlighted here.

### Acknowledgements

This work was supported by the U. S. Army Research Laboratory and the U. S. Army Research Office under Grant W911NF2010219, Office of Naval Research under Grant N000142412673, and NSF.

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

## A. Proof of Lemma 3.2

*Proof.* Since the Mangasarian-Fromovitz constraint qualification (MFCQ) satisfied at any point in the feasible set of problem 1b, there exists a slack variable $\mu \in \mathbb{R}^d$ such that any local maximizer point $\Delta$ of problem 1b has to satisfy the KKT conditions:

$$\nabla l(\theta + \Delta) - 2\mu\Delta = 0, \tag{14a}$$

$$\|\Delta\|_2^2 \leq \rho^2, \tag{14b}$$

$$\mu \geq 0, \tag{14c}$$

$$\mu(\|\Delta\|_2^2 - \rho^2) = 0. \tag{14d}$$

The solution is either $\mu = 0$ or

$$\mu = \frac{\|\nabla l(\theta + \Delta)\|_2}{2\|\Delta\|_2}. \tag{15}$$

By taking the solution of $\mu$ into (14a), we will conclude the necessary conditions 2. $\qquad\square$

## B. Derivation of the Effective Inverse Hessian via KKT Conditions

In this section, we provide a rigorous derivation of the effective inverse Hessian term used in our bilevel update rule. We show that the term $\frac{\rho}{\|\nabla l\|_2} I$ arises naturally from differentiating the KKT conditions of the inner constrained maximization problem under the assumption that the constraint curvature dominates the loss landscape curvature.

### B.1. Problem Setup and KKT Conditions

Consider the inner maximization problem with respect to the perturbation $\Delta$:

$$\max_{\Delta} l(\theta + \Delta) \quad \text{s.t.} \quad \|\Delta\|_2^2 = \rho^2 \tag{16}$$

We define the Lagrangian $\mathcal{L}(\Delta, \lambda)$ for the equivalent minimization problem of $-l(\theta + \Delta)$:

$$\mathcal{L}(\Delta, \lambda) = -l(\theta + \Delta) + \frac{\lambda}{2} \left(\|\Delta\|_2^2 - \rho^2\right) \tag{17}$$

where $\lambda$ is the Lagrange multiplier associated with the spherical constraint. The first-order Karush-Kuhn-Tucker (KKT) conditions for optimality are:

$$\nabla_{\Delta}\mathcal{L} = -\nabla l(\theta + \Delta) + \lambda\Delta = 0 \tag{18a}$$

$$\nabla_{\lambda}\mathcal{L} = \frac{1}{2}(\|\Delta\|_2^2 - \rho^2) = 0 \tag{18b}$$

From the stationarity condition (18a), we have $\lambda\Delta = \nabla l(\theta + \Delta)$. Taking the norm of both sides and using $\|\Delta\|_2 = \rho$, we find the value of the Lagrange multiplier:

$$\lambda = \frac{\|\nabla l(\theta + \Delta)\|_2}{\rho} \tag{19}$$

### B.2. Implicit Differentiation of the KKT System

To find the sensitivity of the optimal perturbation $\Delta^*$ with respect to the model parameters $\theta$, we apply the Implicit Function Theorem by taking the total differential of the KKT conditions with respect to $\theta$, $\Delta$, and $\lambda$.

Differentiating the stationarity condition (18a):

$$-\nabla^2 l(\theta + \Delta)(d\theta + d\Delta) + d\lambda\Delta + \lambda d\Delta = 0 \tag{20}$$

Rearranging to group the unknowns $d\Delta$ and $d\lambda$ on the left-hand side:

$$(\lambda I - \nabla^2 l(\theta + \Delta))d\Delta + \Delta d\lambda = \nabla^2 l(\theta + \Delta)d\theta \tag{21}$$

Differentiating the feasibility condition (18b):

$$\Delta^\top d\Delta = 0 \tag{22}$$

We can write equations (21) and (22) as a block matrix linear system, often referred to as the KKT Matrix system:

$$\underbrace{\begin{bmatrix} \lambda I - \nabla^2 l & \Delta \\ \Delta^\top & 0 \end{bmatrix}}_{\mathbf{K}} \begin{bmatrix} d\Delta \\ d\lambda \end{bmatrix} = \begin{bmatrix} \nabla^2 l \\ 0 \end{bmatrix} d\theta \tag{23}$$

### B.3. The Stiff Constraint Assumption

Solving (23) exactly yields the precise sensitivity $\frac{\partial \Delta}{\partial \theta}$. However, to derive the interpretable form used in our algorithm, we introduce the **Stiff Constraint Assumption**. We assume that the curvature induced by the spherical constraint dominates the local curvature of the loss function:

$$\lambda I \gg \nabla^2 l(\theta + \Delta) \tag{24}$$

Physically, this implies that the "force" keeping the perturbation on the sphere is much stronger than the Hessian of the loss. Under this assumption, we approximate the top-left block of the KKT matrix:

$$\lambda I - \nabla^2 l(\theta + \Delta) \approx \lambda I \tag{25}$$

The simplified KKT system becomes:

$$\begin{bmatrix} \lambda I & \Delta \\ \Delta^\top & 0 \end{bmatrix} \begin{bmatrix} d\Delta \\ d\lambda \end{bmatrix} = \begin{bmatrix} \nabla^2 l \\ 0 \end{bmatrix} d\theta \tag{26}$$

### B.4. Solving via Block Matrix Inversion

We solve for $d\Delta$ by computing the inverse of the block matrix $\mathbf{K} \approx \begin{bmatrix} A & B \\ B^\top & 0 \end{bmatrix}$ where $A = \lambda I$ and $B = \Delta$. Using the formula for block matrix inversion, the top-left block of the inverse $\mathbf{K}_{11}^{-1}$ is given by:

$$\mathbf{K}_{11}^{-1} = A^{-1} - A^{-1}B(B^\top A^{-1} B)^{-1} B^\top A^{-1} \tag{27}$$

Substituting $A^{-1} = \frac{1}{\lambda} I$:

$$\mathbf{K}_{11}^{-1} = \frac{1}{\lambda} I - \frac{1}{\lambda} \Delta \left( \Delta^\top \frac{1}{\lambda} I \Delta \right)^{-1} \Delta^\top \frac{1}{\lambda} \tag{28}$$

$$= \frac{1}{\lambda} \left( I - \frac{\Delta \Delta^\top}{\Delta^\top \Delta} \right) \tag{29}$$

Using the Jacobian relation $d\Delta = \mathbf{K}_{11}^{-1}(\nabla^2 l)d\theta$ derived from (26), we obtain the sensitivity:

$$\frac{\partial \Delta}{\partial \theta} \approx \frac{1}{\lambda} \left( I - \frac{\Delta \Delta^\top}{\|\Delta\|_2^2} \right) \nabla^2 l(\theta + \Delta) \tag{30}$$

### B.5. Recovering the Effective Inverse Hessian

Finally, we substitute the physical values from the stationarity condition back into the equation. Recall from Eq. (19) that $\lambda = \frac{\|\nabla l\|_2}{\rho}$, and that the optimal perturbation is aligned with the gradient, $\Delta = \rho \frac{\nabla l}{\|\nabla l\|_2}$. Substituting these into the sensitivity formula:

$$\frac{\partial \Delta}{\partial \theta} \approx \underbrace{\frac{\rho}{\|\nabla l(\theta + \Delta)\|_2}}_{\text{Scalar Stiffness } \mathcal{H}_{eff}^{-1}} \underbrace{\left( I - \frac{\nabla l \nabla l^\top}{\|\nabla l\|_2^2} \right)}_{\text{Projection } P_{\nabla l}^\perp} \nabla^2 l(\theta + \Delta) \tag{31}$$

This derivation confirms that the term $\frac{\rho}{\|\nabla l\|_2} I$ acts as the effective inverse Hessian in the bilevel update rule. It represents the inverse of the constraint's curvature, projected onto the tangent space of the sphere.

**B.6. Optimal Gradient $\nabla_\theta \Delta^*(\theta)$**

**Lemma B.1.** *For every $(\theta, \Delta) \in R$, it holds that*

$$\nabla_\theta \Delta^*(\theta) = \left(I - \rho \frac{P_{\nabla l^*}^\perp \nabla^2 l^*}{\|\nabla l^*\|}\right)^{-1} \cdot \rho \frac{P_{\nabla l^*}^\perp}{\|\nabla l^*\|} \nabla^2 l^* \tag{32}$$

*Proof.* Define the Jacobian matrix $\nabla_\theta \Delta^*(\theta)$ to be

$$\nabla_\theta \Delta^*(\theta) := \begin{bmatrix} \frac{\partial}{\partial \theta_1} \Delta_1^*(\theta) & \cdots & \frac{\partial}{\partial \theta_d} \Delta_1^*(\theta) \\ \cdots & \cdots & \cdots \\ \frac{\partial}{\partial \theta_1} \Delta_d^*(\theta) & \cdots & \frac{\partial}{\partial \theta_d} \Delta_d^*(\theta) \end{bmatrix}. \tag{33}$$

By the chain rule, it follows that

$$\nabla_\theta L(\theta) = (I + \nabla_\theta \Delta^*(\theta))^\top \nabla l \left(\theta + \Delta^*(\theta)\right) \tag{34}$$

Based on the optimality condition 2a: $\Delta^*(\theta) = \rho \frac{\nabla l(\theta + \Delta^*(\theta))}{\|\nabla l(\theta + \Delta^*(\theta))\|_2}$, we could implicitly derive the derivative for the the optimal $\Delta^*(\theta)$, $l(\theta + \Delta^*(\theta)) = L(\theta)$ is abbreviated by $l^*$ for simplicity.

$$\nabla_\theta \Delta^*(\theta) = \rho \left[\frac{I - \frac{\nabla l^* \nabla l^T}{\|\nabla l^*\|^2}}{\|\nabla l^*\|} \nabla^2 l^*\right] (I + \nabla_\theta \Delta^*(\theta)) \tag{35}$$

Since $I - \rho \frac{\left(I - \frac{\nabla l \nabla l^T}{\|\nabla l\|^2}\right) \nabla^2 l}{\|\nabla l\|}$ is invertible in $R$, we could have the following formula

$$\nabla_\theta \Delta^*(\theta) = \left(I - \rho \frac{P_{\nabla l^*}^\perp \nabla^2 l^*}{\|\nabla l^*\|}\right)^{-1} \cdot \rho \frac{P_{\nabla l^*}^\perp}{\|\nabla l^*\|} \nabla^2 l^* = (I - \rho H^*)^{-1} \rho H^* \tag{36}$$

where $H$ is defined as $H = \frac{P_{\nabla l}^\perp}{\|\nabla l\|} \nabla^2 l$, $P_{\nabla l}^\perp = I - \frac{\nabla l \nabla l^\top}{\|\nabla l\|_2^2}$, and $H^* = \frac{P_{\nabla l^*}^\perp}{\|\nabla l^*\|} \nabla^2 l^*$, $P_{\nabla l^*}^\perp = I - \frac{\nabla l^* \nabla l^{*\top}}{\|\nabla l^*\|_2^2}$. $l(\theta + \Delta)$ is abbreviated by $l$ for simplicity. $\square$

## C. Proof of Theorem 3.3

*Proof.* We analyze the stability of the coupled dynamical system defined in Eq. (9). We define the domain of interest $R$ as the region where the curvature is bounded relative to the gradient magnitude:

$$R = \left\{(\theta, \Delta) \mid \rho \cdot \sigma_1(\nabla^2 l(\theta + \Delta)) < \|\nabla l(\theta + \Delta)\|_2\right\}$$

Consider the following Lyapunov function candidate, which combines the loss value with the distance of the current perturbation $\Delta$ from its ideal direction:

$$V(\theta, \Delta) = l(\theta + \Delta) + \frac{1}{2} \left\|\Delta - \rho \frac{\nabla l(\theta + \Delta)}{\|\nabla l(\theta + \Delta)\|_2}\right\|_2^2 \tag{37}$$

For brevity, let $g = \nabla l(\theta + \Delta)$, $H = \nabla^2 l(\theta + \Delta)$, and let $\mathcal{E}$ represent the perturbation error vector:

$$\mathcal{E} = \Delta - \rho \frac{g}{\|g\|_2}$$

Differentiation of $V$ with respect to time $t$ yields:

$$\frac{dV}{dt} = \nabla l(\theta + \Delta)^\top \left(\frac{d\theta}{dt} + \frac{d\Delta}{dt}\right) + \mathcal{E}^\top \frac{d}{dt} \left(\Delta - \rho \frac{g}{\|g\|_2}\right)$$

$$= g^\top \left(\frac{d\theta}{dt} + \frac{d\Delta}{dt}\right) + \mathcal{E}^\top \left[\frac{d\Delta}{dt} - \rho \frac{P_g^\perp}{\|g\|_2} H \left(\frac{d\theta}{dt} + \frac{d\Delta}{dt}\right)\right]$$

where $P_g^\perp = I - \frac{gg^\top}{\|g\|^2}$ is the projection onto the tangent space of the gradient.

We regroup the terms by the time derivatives $\frac{d\theta}{dt}$ and $\frac{d\Delta}{dt}$:

$$\frac{dV}{dt} = \underbrace{\left[g^\top - \rho\mathcal{E}^\top \frac{P_g^\perp}{\|g\|_2} H\right]}_{\text{Term } A} \frac{d\theta}{dt} + \underbrace{\left[g^\top + \mathcal{E}^\top \left(I - \rho\frac{P_g^\perp}{\|g\|_2}H\right)\right]}_{\text{Term } B} \frac{d\Delta}{dt} \tag{38}$$

Substituting the system dynamics from Eq. (9), specifically $\frac{d\theta}{dt} = -\alpha g$ and $\frac{d\Delta}{dt} = \beta P_\Delta^\perp(\rho\frac{g}{\|g\|_2} - \Delta) = -\beta P_\Delta^\perp \mathcal{E}$, we obtain:

$$\frac{dV}{dt} = -\alpha \left[g^\top - \rho\mathcal{E}^\top \frac{P_g^\perp}{\|g\|_2}H\right]g - \beta\left[g^\top + \mathcal{E}^\top\left(I - \rho\frac{P_g^\perp}{\|g\|_2}H\right)\right]P_\Delta^\perp\mathcal{E} \tag{39}$$

We simplify the terms individually. For the first term (associated with $\alpha$):

- $g^\top g = \|g\|_2^2$.

- The cross term involves $P_g^\perp g = 0$, so the term $\rho\mathcal{E}^\top \frac{P_g^\perp}{\|g\|_2}Hg$ remains.

For the second term (associated with $\beta$):

- Note that $\mathcal{E} = \Delta - \rho\frac{g}{\|g\|_2}$. Since $\Delta$ is on the sphere (or converges to it), $P_\Delta^\perp \Delta = 0$. Thus, $P_\Delta^\perp \mathcal{E} = P_\Delta^\perp(-\rho\frac{g}{\|g\|_2})$.

- The term $g^\top P_\Delta^\perp \mathcal{E} = -\rho g^\top P_\Delta^\perp \frac{g}{\|g\|_2} = -\frac{\rho}{\|g\|_2}g^\top P_\Delta^\perp g$. Since $P_\Delta^\perp$ is positive semi-definite, this contribution is non-positive.

Substituting these simplifications back, we derive an upper bound for $\frac{dV}{dt}$. We use the spectral norm $\sigma_1(H)$ for the cross terms. For the third term, we utilize the geometric property that $g^\top P_\Delta^\perp \mathcal{E} = -\frac{\rho}{\|g\|_2}\|P_\Delta^\perp g\|_2^2$, which allows us to bound its magnitude by $\beta\rho\|g\|_2$.

$$\frac{dV}{dt} = -\alpha\|g\|_2^2 + \alpha\rho\mathcal{E}^\top \frac{P_g^\perp}{\|g\|_2}Hg - \beta g^\top P_\Delta^\perp \mathcal{E} - \beta\mathcal{E}^\top P_\Delta^\perp \mathcal{E} + \beta\rho\mathcal{E}^\top \frac{P_g^\perp}{\|g\|_2}HP_\Delta^\perp \mathcal{E}$$

$$\le -\alpha\|g\|_2^2 + \alpha\rho\|\mathcal{E}\|_2\sigma_1(H) + \beta\rho\|g\|_2 + \frac{\beta\rho\|\mathcal{E}\|_2^2}{\|g\|_2}\sigma_1(H)$$

To guarantee stability ($\frac{dV}{dt} \le 0$), we group the terms scaling with $\|g\|_2$. We require the negative gradient descent term $-\alpha\|g\|_2^2$ to dominate the destabilizing terms:

$$\frac{dV}{dt} \le -\|g\|_2\left(\alpha\|g\|_2 - \beta\rho - \alpha\rho\frac{\|\mathcal{E}\|_2\sigma_1(H)}{\|g\|_2} - \frac{\beta\rho\|\mathcal{E}\|_2^2}{\|g\|_2^2}\sigma_1(H)\right)$$

Thus, a sufficient condition for stability is that the term in the parentheses remains positive. We simplify this condition by applying the triangle inequality bound on the error vector, $\|\mathcal{E}\|_2 \le \|\Delta\|_2 + \rho = 2\rho$. Substituting $\|\mathcal{E}\|_2 \le 2\rho$ into the expression:

$$\beta\rho + \rho\sigma_1(H)\left(\alpha\frac{2\rho}{\|g\|_2} + \beta\frac{4\rho^2}{\|g\|_2^2}\right) < \alpha\|g\|_2 \tag{40}$$

Dividing by $\alpha$ and rearranging to isolate the gradient magnitude:

$$\frac{\beta}{\alpha}\rho + 2\rho^2\frac{\sigma_1(H)}{\|g\|_2}\left(1 + 2\frac{\beta}{\alpha}\frac{\rho}{\|g\|_2}\right) < \|g\|_2 \tag{41}$$

Recall that $H = \frac{P_g^\perp}{\|g\|} \nabla^2 l$, the inequality can be further simplified as:

$$\|g\|_2 > \frac{\beta}{\alpha}\rho + 2\sigma_1(\nabla^2 l)\frac{\rho^2}{\|g\|_2^2} + 4\frac{\beta}{\alpha}\sigma_1(\nabla^2 l)\frac{\rho^3}{\|g\|_2^3} \tag{42}$$

This yields a clear interpretable condition: stability is guaranteed as long as the gradient $\|g\|_2$ is not vanishingly small relative to the perturbation radius $\rho$ and curvature $\sigma_1(\nabla^2 l)$ of each current position of the trajectory. Specifically, for a fixed ratio $\beta/\alpha$, the condition holds whenever $\|g\|_2 > \mathcal{O}(\rho)$.

Finally, $\frac{dV}{dt} = 0$ implies $\|g\|_2 = 0$ (stationary point) and $\|\mathcal{E}\|_2 = 0$ (optimal perturbation), which corresponds to the unique minimax equilibrium:

$$\nabla L(\theta) = 0, \quad \Delta = \rho\frac{\nabla l(\theta + \Delta)}{\|\nabla l(\theta + \Delta)\|_2} = \Delta^*(\theta)$$

$\square$

# D. Proof of Theorem 3.6

*Proof.* We analyze the MSAM update with finite $M$ as a *perturbed* Forward Euler discretization of the ideal ODE. We utilize Assumption 3.4 (Lipschitz continuity) to bound the error propagation and Assumption 3.5 (Bounded Derivatives) to bound the truncation error.

Let $F(\theta) = -\nabla l(\theta + \Delta^*(\theta))$ be the ideal vector field where $\Delta^*(\theta)$ is the exact fixed point. The ideal Euler step is:

$$\tilde{\theta}_{k+1} = \theta_k + \eta F(\theta_k) \tag{43}$$

The actual MSAM update uses an approximation $\Delta_M$ obtained after $M$ fixed-point iterations:

$$\theta_{k+1} = \theta_k - \eta\nabla l(\theta_k + \Delta_M(\theta_k)) \tag{44}$$

We can rewrite the actual update as a perturbed Euler step:

$$\theta_{k+1} = \theta_k + \eta\left(F(\theta_k) + \mathcal{E}_{fp}^{(k)}\right) \tag{45}$$

where $\mathcal{E}_{fp}^{(k)} = \nabla l(\theta_k + \Delta^*) - \nabla l(\theta_k + \Delta_M)$ represents the Fixed-Point Approximation Error vector at step $k$.

First, we bound the error in the perturbation $\Delta$. Since the inner map $T_\theta(\Delta)$ is a contraction with rate $\kappa < 1$ (implied by the curvature bound relative to the gradient in the region $R$), the distance to the fixed point shrinks by $\kappa$ at every inner iteration:

$$\|\Delta_M - \Delta^*\| \leq \kappa^M\|\Delta_0 - \Delta^*\| \leq \kappa^M(2\rho)$$

Since $\nabla l$ is Lipschitz continuous with constant $L$ (Assumption 3.4):

$$\|\mathcal{E}_{fp}^{(k)}\| = \|\nabla l(\theta_k + \Delta^*) - \nabla l(\theta_k + \Delta_M)\| \leq L\|\Delta^* - \Delta_M\| \leq 2\rho L\kappa^M$$

Thus, the vector field perturbation is uniformly bounded by $B_{fp} \triangleq 2\rho L\kappa^M$.

We define the global error at step $k$ as $e_k = \|\theta_k - \theta(t_k)\|$. To derive the recurrence for $e_{k+1}$, we explicitly compare the discrete update rule with the Taylor expansion of the continuous solution.

We expand the true solution $\theta(t)$ around $t_k$ using Taylor's Theorem with Lagrange remainder. Noting that $t_{k+1} = t_k + \eta$, we have:

$$\theta(t_{k+1}) = \theta(t_k) + \eta\frac{d\theta}{dt}\bigg|_{t_k} + \frac{\eta^2}{2}\frac{d^2\theta}{dt^2}\bigg|_{\xi_k}$$
$$= \theta(t_k) + \eta F(\theta(t_k)) + \mathcal{R}_k$$

where $\xi_k \in [t_k, t_{k+1}]$. The Local Truncation Error is $\mathcal{R}_k = \frac{\eta^2}{2}\ddot{\theta}(\xi_k)$. To bound this, we expand $\ddot{\theta}(t) = \frac{d}{dt}F(\theta(t)) = \nabla F(\theta) \cdot F(\theta)$. Using Assumption 3.5, the derivatives $\nabla l$ and $\nabla^2 l$ are bounded by $B_1$ and $B_2$. By the Implicit Function Theorem, $\nabla \Delta^*(\theta)$ is bounded. Therefore, $\|\ddot{\theta}(t)\|$ is bounded by a constant $M_{ode}$ depending on $B_1, B_2$:

$$\|\mathcal{R}_k\| \leq \frac{1}{2}M_{ode}\eta^2$$

Subtracting the continuous expansion from the discrete update:

$$\theta_{k+1} - \theta(t_{k+1}) = \left[\theta_k + \eta F(\theta_k) + \eta \mathcal{E}_{fp}^{(k)}\right] - [\theta(t_k) + \eta F(\theta(t_k)) + \mathcal{R}_k]$$

$$= (\theta_k - \theta(t_k)) + \eta(F(\theta_k) - F(\theta(t_k))) + \eta \mathcal{E}_{fp}^{(k)} - \mathcal{R}_k$$

Taking the Euclidean norm and applying the triangle inequality:

$$e_{k+1} \leq \|\theta_k - \theta(t_k)\| + \eta\|F(\theta_k) - F(\theta(t_k))\| + \eta\|\mathcal{E}_{fp}^{(k)}\| + \|\mathcal{R}_k\|$$

Since $\nabla^2 l$ is bounded (Assumption 3.5), the vector field $F(\theta)$ is Lipschitz continuous with constant $L_F$. Thus:

$$\|F(\theta_k) - F(\theta(t_k))\| \leq L_F\|\theta_k - \theta(t_k)\| = L_F e_k$$

Substituting the bounds:

$$e_{k+1} \leq (1 + \eta L_F)e_k + \underbrace{\frac{1}{2}M_{ode}\eta^2}_{\text{Discretization Error}} + \underbrace{\eta B_{fp}}_{\text{Fixed-Point Error}} \tag{46}$$

Iterating this recurrence yields:

$$e_N \leq \left(\frac{M_{ode}\eta}{2} + B_{fp}\right)\sum_{j=0}^{N-1}(1 + \eta L_F)^j$$

Summing the geometric series $\sum_{j=0}^{N-1} r^j = \frac{r^N - 1}{r-1}$ where $r = 1 + \eta L_F$:

$$e_N \leq \left(\frac{M_{ode}\eta}{2} + 2\rho L\kappa^M\right)\frac{(1 + \eta L_F)^N - 1}{\eta L_F}$$

Using the bound $(1 + x)^N \leq e^{Nx}$ and substituting $N\eta = T$:

$$e_N \leq \left(\frac{M_{ode}\eta}{2} + 2\rho L\kappa^M\right)\frac{e^{L_F T} - 1}{\eta L_F}$$

Simplifying the expression to separate the terms:

$$e_N \leq \underbrace{\left[\frac{M_{ode}}{2L_F}(e^{L_F T} - 1)\right]}_{C_1}\eta + \underbrace{\left[\frac{2\rho L}{L_F \eta}(e^{L_F T} - 1) \cdot \eta\right]}_{C_2}\kappa^M$$

Canceling $\eta$ in the second term ensures the fixed-point error depends only on $M$:

$$\sup_{0 \leq k \leq N}\|\theta_k - \theta(t_k)\| \leq C_1\eta + C_2\kappa^M$$

$\square$

The constants $L_F$ and $M_{ode}$ governing the error bounds in Theorem 3.6 are not arbitrary; they are determined by the local geometry of the loss landscape. Under Assumption 3.5, and letting $c = \inf_{w \in R}\|\nabla l(w)\|$ be the minimum gradient norm, explicit upper bounds are given by:

$$L_F \leq \frac{B_2}{1 - \kappa}, \quad M_{ode} \leq \frac{B_1 B_2}{1 - \kappa} \tag{47}$$

where $\kappa \triangleq \frac{\rho B_2}{c}$ is the contraction ratio of the inner fixed-point map.

## D.1. Derivation of Stability Constants

In this part, we derive the explicit upper bounds for the Lipschitz constant $L_F$ and the trajectory acceleration bound $M_{ode}$. These bounds rely on the implicit sensitivity of the worst-case perturbation.

### D.1.1. SENSITIVITY OF THE OPTIMAL PERTURBATION

Recall that the optimal perturbation $\Delta^*(\theta)$ is the fixed point of the mapping $T(\theta, \Delta) = \rho \frac{\nabla l(\theta + \Delta)}{\|\nabla l(\theta + \Delta)\|_2}$. Let $u = \theta + \Delta$. The Jacobian of the map $T$ with respect to $u$ is given by:

$$J_T(u) = \rho \frac{\|\nabla l\|^2 I - \nabla l \nabla l^\top}{\|\nabla l\|^3} \nabla^2 l(u) = \rho \frac{P_{\nabla l}^\perp}{\|\nabla l\|} \nabla^2 l(u)$$

Using Assumption 3.5, we bound the spectral norm of this Jacobian:

$$\|J_T(u)\|_2 \leq \frac{\rho}{\|\nabla l\|} \|\nabla^2 l\| \leq \frac{\rho B_2}{c} \triangleq \kappa$$

By the Implicit Function Theorem, differentiating the fixed-point equation $\Delta^*(\theta) = T(\theta, \Delta^*(\theta))$ with respect to $\theta$ yields:

$$\nabla_\theta \Delta^* = \nabla_\theta T + \nabla_\Delta T (\nabla_\theta \Delta^*)$$

Since $T$ depends on the sum $\theta + \Delta$, we have $\nabla_\theta T = \nabla_\Delta T = J_T$. Thus:

$$(I - J_T) \nabla_\theta \Delta^* = J_T \implies \nabla_\theta \Delta^* = (I - J_T)^{-1} J_T$$

Taking norms and using the contraction property $\|J_T\| \leq \kappa < 1$:

$$\|\nabla_\theta \Delta^*\| \leq \frac{\|J_T\|}{1 - \|J_T\|} \leq \frac{\kappa}{1 - \kappa} \tag{48}$$

### D.1.2. LIPSCHITZ CONSTANT OF THE VECTOR FIELD ($L_F$)

The effective vector field is $F(\theta) = -\nabla l(\theta + \Delta^*(\theta))$. Differentiating with respect to $\theta$:

$$\nabla_\theta F(\theta) = -\nabla^2 l(\theta + \Delta^*)(I + \nabla_\theta \Delta^*)$$

Taking the spectral norm:

$$\|\nabla_\theta F(\theta)\| \leq \|\nabla^2 l\|(1 + \|\nabla_\theta \Delta^*\|) \leq B_2 \left(1 + \frac{\kappa}{1 - \kappa}\right) = B_2 \left(\frac{1}{1 - \kappa}\right)$$

Thus, we establish the bound:

$$L_F \leq \frac{B_2}{1 - \kappa} \tag{49}$$

### D.1.3. TRAJECTORY ACCELERATION BOUND ($M_{ode}$)

The acceleration of the continuous trajectory is given by the total derivative of the velocity field:

$$\ddot{\theta}(t) = \frac{d}{dt} F(\theta(t)) = \nabla_\theta F(\theta) \cdot \frac{d\theta}{dt} = \nabla_\theta F(\theta) \cdot F(\theta)$$

We apply the Cauchy-Schwarz inequality and substitute the previous bounds. We use the gradient bound $\|F(\theta)\| = \|\nabla l(\theta + \Delta^*)\| \leq B_1$:

$$\|\ddot{\theta}(t)\| \leq \|\nabla_\theta F(\theta)\| \|F(\theta)\| \leq \left(\frac{B_2}{1 - \kappa}\right) B_1$$

Thus, the trajectory smoothness constant is bounded by:

$$M_{ode} \leq \frac{B_1 B_2}{1 - \kappa} \tag{50}$$

# E. Linear Stability of the Unprojected Flow

Without projecting the inner maximiser onto the $\rho$-sphere, the continuous-time dynamics is

$$\begin{cases} \dfrac{d\Delta}{dt} = \rho \cdot \dfrac{\nabla L(\theta + \Delta)}{\|\nabla L(\theta + \Delta)\|_2} - \Delta \\ \dfrac{d\theta}{dt} = -(I + \dfrac{d\Delta}{d\theta})^\top \nabla L(\theta + \Delta) \end{cases}$$

$$\frac{d}{dt} \begin{bmatrix} \Delta \\ \theta \end{bmatrix} \approx J \begin{bmatrix} \Delta \\ \theta \end{bmatrix},$$

The Jacobian $J$ is derived by differentiating the ODEs with respect to $\Delta$ and $\theta$, at $x = \theta + \Delta$:

$$J = \begin{bmatrix} \frac{\partial \dot\Delta}{\partial \Delta} & \frac{\partial \dot\Delta}{\partial \theta} \\ \frac{\partial \dot\theta}{\partial \Delta} & \frac{\partial \dot\theta}{\partial \theta} \end{bmatrix}.$$

Term 1: $\frac{\partial \dot\Delta}{\partial \Delta}$

$$\frac{\partial \dot\Delta}{\partial \Delta} = \frac{\rho}{\|\nabla L(x)\|_2} \left[ I - \frac{\nabla L(x)\nabla L(x)^\top}{\|\nabla L(x)\|_2^2} \right] \nabla^2 L(x) - I$$

Term 2: $\frac{\partial \dot\Delta}{\partial \theta}$

$$\frac{\partial \dot\Delta}{\partial \theta} = \frac{\rho}{\|\nabla L(x)\|_2} \left[ I - \frac{\nabla L(x)\nabla L(x)^\top}{\|\nabla L(x)\|_2^2} \right] \nabla^2 L(x)$$

Term 3: $\frac{\partial \dot\theta}{\partial \Delta}$ and $\frac{\partial \dot\theta}{\partial \theta}$

Define $A = \frac{I - \frac{\nabla L \nabla L^\top}{\|\nabla L\|^2}}{\|\nabla L\|} \nabla^2 L$, we could write the following first order approximation:

$$\frac{d}{d\Delta}\left( \frac{d\theta}{dt} \right) \approx -\frac{d}{d\Delta}[(I + \rho A)\nabla L(\theta + \Delta)]$$

$$= -\rho \frac{dA}{d\Delta} \nabla L - (I + \rho A)\nabla^2 L$$

$$\frac{d}{d\theta}\left( \frac{d\theta}{dt} \right) \approx -\rho \frac{dA}{d\theta} \nabla L - (I + \rho A)(I + \rho A)\nabla^2 L$$

$$\approx -\rho(I + \rho A)\frac{dA}{d\Delta} \nabla L - (2\rho A + I)\nabla^2 L$$

$$\frac{dA}{d\Delta} = -\frac{\nabla^2 L \nabla L \nabla L^T \nabla^2 L}{\|\nabla L\|^5} + \frac{\nabla^2 L}{\|\nabla L\|^3} + \frac{I - \frac{\nabla L \nabla L^\top}{\|\nabla L\|^2}}{\|\nabla L\|} \nabla^3 L$$

Hence, the first order approximation is

$$J \approx \begin{bmatrix} \frac{\rho}{\|\nabla L(x)\|_2} \left[ I - \frac{\nabla L(x)\nabla L(x)^\top}{\|\nabla L(x)\|_2^2} \right] \nabla^2 L(x) - I & \frac{\rho}{\|\nabla L(x)\|_2} \left[ I - \frac{\nabla L(x)\nabla L(x)^\top}{\|\nabla L(x)\|_2^2} \right] \nabla^2 L(x) \\ -\rho \frac{dA}{d\Delta} \nabla L - (I + \rho A) \nabla^2 L & -\rho \frac{dA}{d\Delta} \nabla L - (2\rho A + I)\nabla^2 L \end{bmatrix}.$$

For the unperturbed system ($\rho = 0$), the Jacobian matrix simplifies to:

$$J_0 = \begin{bmatrix} -I & 0 \\ -H & -H \end{bmatrix}.$$

For the $-I$ block: $\mu = -1$ (with multiplicity equal to the dimension of $\Delta$). For the $-H$ block: $\mu = -\lambda_i$, where $\lambda_i > 0$ are eigenvalues of $H$. All eigenvalues have negative real parts; the equilibrium is locally asymptotically stable.

For the perturbed system ($\rho \ll 1$), the Jacobian becomes $J = J_0 + \rho J_1 + \mathcal{O}(\rho^2)$, where:

$$J_1 = \begin{bmatrix} \frac{1}{\|\nabla L\|_2} \left[ I - \frac{\nabla L \nabla L^\top}{\|\nabla L\|_2^2} \right] H & \frac{1}{\|\nabla L\|_2} \left[ I - \frac{\nabla L \nabla L^\top}{\|\nabla L\|_2^2} \right] H \\ -\frac{dA}{d\Delta} \nabla L - A \nabla^2 L & -\frac{dA}{d\Delta} \nabla L - 2A \nabla^2 L \end{bmatrix}.$$

When it comes to the leading-order eigenvalue perturbations, for small $\rho$, eigenvalues $\mu$ of $J$ are perturbed from their $\rho = 0$ values. Using first-order perturbation theory:

$$\mu = \mu_0 + \rho \mu_1 + \mathcal{O}(\rho^2),$$

where $\mu_0$ are eigenvalues of $J_0$, and $\mu_1 = \frac{\mathbf{v}_0^\top J_1 \mathbf{u}_0}{\mathbf{v}_0^\top \mathbf{u}_0}$, with $\mathbf{u}_0, \mathbf{v}_0$ being the right/left eigenvectors of $J_0$. For each eigenvalue $\mu_0 = -1$ (from $-I$): its right eigenvector is $\mathbf{u}_0 = \begin{bmatrix} \mathbf{a} \\ 0 \end{bmatrix}$, where $\mathbf{a}$ is arbitrary, and its left eigenvector is $\mathbf{v}_0 = \begin{bmatrix} \mathbf{a} \\ 0 \end{bmatrix}$. For eigenvalues $\mu_0 = -\lambda_i$ (from $-H$), its right eigenvector is $\mathbf{u}_0 = \begin{bmatrix} 0 \\ \mathbf{b}_i \end{bmatrix}$, where $\mathbf{b}_i$ is the eigenvector of $H$, and its left eigenvector is $\mathbf{v}_0 = \begin{bmatrix} 0 \\ \mathbf{b}_i \end{bmatrix}$.

For the first-order corrections, we could have:

Case 1: Perturbations to $\mu_0 = -1$

For eigenvectors $\mathbf{u}_0 = \begin{bmatrix} \mathbf{a} \\ 0 \end{bmatrix}$, the perturbation term $\mu_1$ is:

$$\mu_1 = \frac{\mathbf{v}_0^\top J_1 \mathbf{u}_0}{\mathbf{v}_0^\top \mathbf{u}_0} = \frac{\mathbf{a}^\top \left( \frac{1}{\|\nabla L\|_2} \left[ I - \frac{\nabla L \nabla L^\top}{\|\nabla L\|_2^2} \right] H \mathbf{a} \right)}{\mathbf{a}^\top \mathbf{a}}.$$

The projection operator $P = \left[ I - \frac{\nabla L \nabla L^\top}{\|\nabla L\|_2^2} \right]$ projects vectors onto the subspace orthogonal to $\nabla L$. Since the eigenvalues of a projection matrix are 0 and 1, $\mu_1$ is upper bounded by the norm of $H$:

$$\mu_1 \leq \frac{1}{\|\nabla L\|_2} \cdot \frac{\mathbf{a}^\top H \mathbf{a}}{\mathbf{a}^\top \mathbf{a}} \leq \frac{\lambda_{\max}(H)}{\|\nabla L\|_2}$$

The term $\mu_1$ reduces the negativity of $\mu$, but stability persists if:

$$\rho \cdot \frac{\lambda_{\max}(H)}{\|\nabla L\|_2} < 1.$$

Case 2: Perturbations to $\mu_0 = -\lambda_i$

For eigenvectors $\mathbf{u}_0 = \begin{bmatrix} 0 \\ \mathbf{b}_i \end{bmatrix}$, the perturbation term $\mu_1$ is:

$$\tilde{\mu}_1 = \frac{\mathbf{v}_0^\top J_1 \mathbf{u}_0}{\mathbf{v}_0^\top \mathbf{u}_0} = \lambda_{\max}(\frac{dA}{d\Delta} \nabla L + 2A \nabla^2 L)$$

Hence, eigenvalues near $\mu_0 = -1$ remain $\mu \approx -1 + \rho \cdot \frac{\lambda_{\max}(H)}{\|\nabla L\|_2} + \mathcal{O}(\rho^2)$. Eigenvalues near $\mu_0 = -\lambda_i$ remain $\mu \approx -\lambda_i + \rho \tilde{\mu}_1 + \mathcal{O}(\rho^2)$. Those eigenvalues remain negative provided $\rho \lambda_{\max}(H) < \|\nabla L\|_2$.

The spectrum of $J_0$ shows that the unperturbed flow is a strict contraction in both $(\Delta, \theta)$ directions. Adding the SAM perturbation introduces off-diagonal couplings of size $\rho$; eigenvalues move toward the imaginary axis but remain in the

left half-plane as long as $\rho$ is smaller than the gradient-to-curvature ratio, which is exactly the same condition guarantees contraction of the fixed-point map used in the main text. Practically, this explains why SAM is stable for moderate radius yet may diverge if $\rho$ is chosen too large relative to the local Hessian. The analysis also highlights that extra inner steps-our MSAM scheme, reduce the effective $\rho$ seen by the outer update, enlarging the stable range without sacrificing robustness.

## F. Additional Experiments

### F.1. Matrix Sensing Problem

**Experimental Setup and Results**: We perform the overparametrized matrix sensing problem using the objective function and test error proposed in (Li et al., 2018) to compare the generalization performance of SAM and MSAM on synthetic data. We generate the true matrix $\mathbf{X}^* = \mathbf{U}^*(\mathbf{U}^*)^\top$ by sampling each entry of $\mathbf{U}^* \in \mathbb{R}^{r \times d}$ from a normal distribution and normalize each column of $\mathbf{U}^*$ to have unit norm. Then we independently sample each entry of the sensing matrix $\mathbf{A_i} \in \mathbb{R}^{d \times d}$ for $i = 1, .., m$ from a normal distribution. We use learning rate $0.0025$ for GD, SAM and MSAM ($M = 3$) and neighborhood sizes $\rho = \{0.001, 0.01, 0.1\}$, $m = 5dr$, $d = 100$, $r = 5$.

**Results and Discussion:** Table 11 and Table 12 show the results of the matrix sensing problem and the MSAM results are comparable to the GD and SAM results when $\rho = 0.001, 0.01$. When $\rho = 0.1$, the larger error of MSAM may be due to the choice of always keeping $\|\Delta\| = \rho$ in solving the inner optimization problem while the true optimal $\Delta$ may have norm less than $\rho$. Similar phenomenon appears in Table 13 where MSAM shifts a distance of $\rho$ from the global optimum.

*Table 11.* Objective Function Value

| $\rho$ | GD | SAM | MSAM (M=3) |
|---|---|---|---|
| 0.001 | 0.5142 | 0.4976 | 0.4976 |
| 0.01 | 0.5142 | 0.4990 | 0.499 |
| 0.05 | 0.5142 | 0.5055 | 0.5056 |
| 0.1 | 0.5142 | 0.5142 | 0.5145 |

*Table 12.* Test Error

| $\rho$ | GD | SAM | MSAM (M=3) |
|---|---|---|---|
| 0.001 | 3.1493 | 3.0922 | 3.0923 |
| 0.01 | 3.1493 | 3.0973 | 3.0973 |
| 0.05 | 3.1493 | 3.1201 | 3.1202 |
| 0.1 | 3.1493 | 3.1493 | 3.1492 |

### F.2. Non-convex Function

**Experimental Setup:** We first verify the convergence of Multi-step Sharpness-Aware Minimization (MSAM) to local stationary points in optimizing the non-convex polynomial function $f_1(\theta) = \theta^2 - \frac{3}{4}\theta^3 + (\theta - 1)^4$. The three local stationary points occur at $\theta_1 = 0.5245$, $\theta_2 = 0.8859$, and $\theta_3 = 2.1521$. The global minimum is $\theta_3$, and $\theta_2$ is a local maximum. We compare GD baseline with SAM ($M = 1$) and MSAM ($M = 3$) with learning rate $0.05$ and neighborhood sizes $\rho \in \{0.01, 0.02, 0.05, 0.1\}$. All algorithms are initialized at $\theta = -1$, and $\Delta = 0$ for MSAM.

**Results and Discussion** Table 13 presents the solution each algorithm outputs. In this simple synthetic experiment, we observe that when the learning rate is small such that the naive gradient descent is trapped in the local minimum, the increment in the neighborhood size $\rho$ helps SAM and MSAM escape the local minima.

*Table 13.* Comparing SAM and MSAM Convergence across $\rho$ for $f_2$

| $\rho$ | GD | Standard SAM | MSAM (M=3) |
|---|---|---|---|
| 0.01 | 0.5252 | 0.5242 | 0.5152 |
| 0.02 | 0.5252 | 0.8844 | 2.1820 |
| 0.05 | 0.5252 | 2.1338 | 2.2020 |
| 0.10 | 0.5252 | 2.1060 | 2.2520 |

## F.3. Spurious Matrix Sensing Partial Results

*Table 14.* Loss for GD, SAM, $n = 10$

| $\rho$ | GD | SAM |
|------|------|------|
| 0.10 | 1.0765e-05 | 1.5702e-05 |
| 0.20 | 1.0765e-05 | 0.50179 |
| 0.30 | 1.0765e-05 | 2.4993 |

*Table 15.* Loss for GD, MSAM, $n = 10$

| $\rho$ | GD | MSAM (M=5) | MSAM (M=10) |
|------|------|------|------|
| 0.10 | 1.0765e-05 | 2.2311e-05 | 3.1253e-05 |
| 0.20 | 1.0765e-05 | 1.2377e-05 | 0.0002 |
| 0.30 | 1.0765e-05 | 2.4993 | 0.00026 |

## F.4. Convex function

**Experimental Setup and Results:** We also verify the convergence of MSAM ($M = 5$, $M = 10$) compared to SAM and Gradient Descent (GD) using $f_1(\theta_1, \theta_2) = 0.25(\theta_1^2 + \theta_2^2) - 0.5\theta_1 + 0.25\theta_2$. The analytical global minima of this function is $(1, -0.5)$. Initialization is $(-1, 0)$ and learning rate is $0.05$ for all algorithms. Table 17 shows that MSAM converges to a neighborhood around the global minima, and, in general, solving the inner optimization for more iterations helps MSAM to converge closer to the global optima.

*Table 16.* GD and SAM Convergence across $\rho$ for $f_1$

| $\rho$ | GD | Standard SAM |
|------|------|------|
| 0.01 | (0.9962, -0.4991) | (1.000, -0.5000) |
| 0.05 | (0.9962, -0.4991) | (0.9994, -0.4998) |
| 0.10 | (0.9962, -0.4991) | (0.9988, -0.4997) |

*Table 17.* MSAM Convergence across $\rho$ for $f_1$

| $\rho$ | MSAM (M=5) | MSAM (M=10) |
|------|------|------|
| 0.01 | (0.9954, -0.4953) | (0.9987, -0.4987) |
| 0.05 | (0.9963, -0.4963) | (1.0044, -0.5054) |
| 0.10 | (1.0340, -0.5340) | (0.9665, -0.4664) |

## F.5. Matrix Completion: Regime-Dependent Tradeoffs

To further investigate the interplay between $\rho$ and $M$, we evaluate MSAM in a controlled matrix completion setting ($n = 10$, $\epsilon = 0.005$). Tables 18 and 19 summarize the final loss achieved across different hyperparameters.

As shown in Table 18, when using a smaller perturbation radius ($\rho = 0.05$), the effect of increasing $M$ is not universally monotone. Moderate inner-loop counts ($M \in \{3, 4\}$) improve the final loss, but excessive inner steps ($M = 5$) begin to weaken convergence accuracy. Conversely, as shown in Table 19, when the perturbation radius is increased to $\rho = 0.06$, larger inner-loop counts become generally more favorable. This empirically supports the regime-dependent tradeoff discussed in our theoretical analysis: larger $\rho$ values require larger $M$ values to accurately resolve the worst-case perturbation and maintain stable convergence.

## F.6. Image Classification

We conducted experiments on the CIFAR-100 dataset using the ResNet-20 architecture. All models were trained using a standard hyperparameter configuration: a fixed batch size of 128, a base learning rate of 0.1, a momentum of 0.9, and a weight decay of $5 \times 10^{-4}$. To account for the stochastic nature of network initialization and mini-batch sampling, each

*Table 18.* Matrix Completion Results ($\rho = 0.05$). At this smaller radius, moderate inner-loop counts improve final loss, while excessive inner steps can weaken convergence accuracy.

| $n$ | $\epsilon$ | Method | $\rho$ | $M$ | Final Loss |
|---|---|---|---|---|---|
| 10 | 0.005 | GD | - | - | $7.8730 \times 10^{-6}$ |
| 10 | 0.005 | SAM | 0.05 | 1 | $9.9114 \times 10^{-6}$ |
| 10 | 0.005 | MSAM | 0.05 | 2 | $8.4769 \times 10^{-6}$ |
| 10 | 0.005 | MSAM | 0.05 | 3 | $6.8657 \times 10^{-6}$ |
| 10 | 0.005 | MSAM | 0.05 | 4 | $7.0932 \times 10^{-6}$ |
| 10 | 0.005 | MSAM | 0.05 | 5 | $1.0412 \times 10^{-5}$ |

*Table 19.* Matrix Completion Results ($\rho = 0.06$). At a larger radius, higher inner-loop counts are required to accurately resolve the worst-case perturbation and achieve favorable convergence.

| $n$ | $\epsilon$ | Method | $\rho$ | $M$ | Final Loss |
|---|---|---|---|---|---|
| 10 | 0.005 | GD | - | - | $1.0581 \times 10^{-5}$ |
| 10 | 0.005 | SAM | 0.06 | 1 | $1.7514 \times 10^{-5}$ |
| 10 | 0.005 | MSAM | 0.06 | 2 | $9.5396 \times 10^{-6}$ |
| 10 | 0.005 | MSAM | 0.06 | 3 | $1.1870 \times 10^{-5}$ |
| 10 | 0.005 | MSAM | 0.06 | 4 | $9.7472 \times 10^{-6}$ |
| 10 | 0.005 | MSAM | 0.06 | 5 | $1.0459 \times 10^{-5}$ |

configuration was executed across five independent runs using distinct random seeds. The final performance metrics are reported as the mean Top-1 validation accuracy alongside the standard deviation across these five runs, providing a statistically robust comparison between standard SGD, SAM, and our MSAM approach.

*Table 20.* Experimental Results on CIFAR-100. The reported accuracy represents the Mean $\pm$ Standard Deviation across 5 random seeds.

| Architecture | Optimizer | Perturbation Radius ($\rho$) | $M$ Steps | Top-1 Accuracy (%) |
|---|---|---|---|---|
| ResNet-20 | SGD | 0.00 | 0 | $68.65 \pm 0.41$ |
| ResNet-20 | SAM | 0.05 | 0 | $69.46 \pm 0.32$ |
| ResNet-20 | MSAM | 0.10 | 5 | $70.01 \pm 0.29$ |

Furthermore, to provide additional empirical context from foundational literature, we will explicitly point to Appendix C.6 of the original SAM paper (Foret et al., 2021). Their ablation study ($\rho = 0.05$ for both SAM and MSAM) independently validates the benefits of taking multiple projected gradient steps for the inner maximization.

Specifically, evaluating a WideResNet architecture on the CIFAR-100 dataset, they demonstrated that increasing the inner steps ($M$) uncovers a significantly larger worst-case perturbation (captured by the "Estimated sharpness" metric), which forces the outer optimizer to penalize sharpness more rigorously and subsequently yields higher generalization accuracy. This historical empirical evidence aligns perfectly with our theoretical framework (Theorem 3.6): taking multiple inner steps ($M > 1$) reduces the truncation error of the worst-case perturbation. By more accurately resolving the true adversarial direction, MSAM enforces a stricter flatness penalty, allowing it to bypass suboptimal local minima and achieve superior test accuracy compared to standard 1-step SAM.

