# OpenReview forum: "Understanding SAM through Minimax Perspective"
_ICML.cc/2026/Conference — ICML 2026 regular_

### Official Review · Reviewer_xSYL · 2026-03-08

**Soundness:** 3
**Presentation:** 2
**Significance:** 2
**Originality:** 3
**Overall Recommendation:** 4
**Confidence:** 3

**Summary:**

This paper introduces a two-timescale ordinary differential equation (ODE) system that models the joint continuous-time dynamics of the model parameters $  \theta  $ and the adversarial perturbation $  \Delta  $ in Sharpness-Aware Minimization (SAM), which addresses the min-max objective $  \min_{\theta} \max_{\|\Delta\| \leq \rho} \ell(\theta + \Delta)  $. The ODE formulation provides a continuous-time lens on SAM's behavior, particularly in non-convex settings, and motivates algorithmic refinements such as multi-step inner maximization (MSAM) to better approximate the worst-case perturbation.

The primary theoretical contributions include:
1. **ODE convergence characterization (Theorem 3.3).**
The authors show that the ODE dynamics converge to a set $\Omega$ where the gradient magnitude of the loss is bounded by the local curvature, providing a characterization of the equilibrium behavior of the SAM dynamics.
2. **Discrete–continuous approximation guarantee (Theorem 3.6).** A uniform error bound  demonstrating that the discrete multi-step SAM (MSAM) trajectory tracks the continuous ODE flow under bounded-curvature assumptions. The approximation error decreases as the number of inner maximization steps $M$ increases.

Based on the characterization in Theorem 3.3, the paper further provides heuristic insights into the role of the inner maximization steps $M$ in MSAM. These theoretical implications are accompanied by empirical evaluations on synthetic objectives and CIFAR-10, intended to support the predicted behavior of the algorithm.

**Compliance With Llm Reviewing Policy:**

Affirmed.

**Final Justification:**

The paper introduces a novel two-timescale ODE framework for Sharpness-Aware Minimization (SAM), providing a continuous-time perspective on its non-convex dynamics and discretization through MSAM.

The rebuttal effectively addressed several concerns. For example, the authors resolved potential division-by-zero issues, and provided additional experiments for intermediate inner steps ($M$). These updates strengthen the paper's foundation and readability. The main remaining weakness is that the mechanistic explanation for SAM’s preference for flat minima and the trade-offs involving the ratio $\beta/\alpha$ remain somewhat informal. While the authors' logic regarding Lyapunov stability and high-curvature repulsion is intuitive, the paper's impact depends on formalizing these claims. I have raised my score based on the authors' commitment to include a formal corollary and a more nuanced discussion of convergence trade-offs in the final version.

**Key Questions For Authors:**

I would appreciate if the authors could address the concerns raised in Point 2 of the Major Weaknesses, and provide further explanations regarding the "Readability and Clarity" issue.

**Limitations:**

Please see the weaknesses part.

**Strengths And Weaknesses:**

# Strengths
The paper proposes an interesting theoretical framework for studying Sharpness-Aware Minimization (SAM) using a two-timescale ODE formulation. The analysis combines continuous-time dynamics with Lyapunov-based stability arguments (Theorem 3.3) and discretization error bounds for the corresponding algorithmic variant MSAM (Theorem 3.6).

This perspective provides a structured way to reason about the interaction between the parameter update and the adversarial perturbation dynamics, and offers a continuous-time interpretation of the algorithm without relying on strong convexity assumptions.

# Major Weaknesses
**1. Unclear connection between the ODE dynamics and SAM’s preference for flat minima.**

The informal explanation in Lines 193--197 ("The adversarial pressure increases Lyapunov energy, effectively bouncing the trajectory out of sharp regions until it finds a basin flat enough to sustain stable convergence.'') is appealing but appears under-supported. Theorem 3.3's inequality prevents divergence or oscillation but does not explicitly exclude the ODE's convergence to sharp (high-curvature) region with small gradients. If the flow can still settle in such sharp regions, this would be inconsistent with SAM's well-documented tendency to favor flatter basins. Stronger bounds or clarification excluding sharp attractors would substantially strengthen the mechanistic explanation of SAM's generalization benefit.

**2. Insufficient support for the claim regarding large** $\beta/\alpha$.

Lines 198–205 suggest that choosing a large ratio $\beta/\alpha$ prevents the algorithm from converging to stationary points with very small gradients. However, this claim does not appear to follow directly from the stated convergence result.

The convergence set $\Omega$ defined in Theorem 3.3 may still contain stationary points with small gradients, and the current analysis does not seem to exclude convergence to such points even when $\beta/\alpha$ is large. As a result, it is unclear whether the theoretical results truly justify the statement that large $\beta/\alpha$ avoids such stationary points.

This issue is important because the paper draws a key practical insight from this claim: that using a large number of inner fixed-point iterations $M$ (which corresponds to a large ratio $\beta/\alpha$) may lead to poor performance of MSAM. This observation is used to explain the empirical phenomenon that increasing $M$ often degrades performance. However, if the theoretical claim does not hold, this explanation is no longer supported by the analysis. In particular, Theorem 3.6 suggests that increasing $M$ improves the approximation of the discrete algorithm to the underlying ODE, which would imply that larger $M$ should be beneficial. This appears inconsistent with the empirical observations reported in the paper (e.g., Table~2), and therefore weakens the soundness of this paper and the proposed explanation for the observed behavior in Point 3 (Line 104-106) of the claimed contributions.

Also, Section 4.1.2 is presented as experimental support for this claim (Lines 209–211), but if the theoretical argument does not actually exclude convergence to small-gradient stationary points under large $M$ values, the interpretation of these experiments becomes less clear.

**3. Experimental evaluation of the effect of** $M$.

In the experiment whose result is reported in Figure 1, the value of $\rho$ is not reported for MSAM. Also, it may be better to perform more experiments with MSAM, where the value of $M$ gradually increases from 1 to 10. Theorem 3.6 can be more clearly verified if the results show a monotone pattern between $M$ and MSAM performance. However, if the results show that as $M$ increases, the algorithm alternates between convergence to a local minimum and inconvergence, then Theorem 3.6 is not verified by this experiment.

**4. Readability and clarity issues.**
 The presentation is dense and at times difficult to follow, with several important claims lacking sufficient justification or clear pointers:

- Lines 189--194: "While in regions of high
curvature that... manifests as transient instability." This statement is made without sufficient justifications. Could the authors clarify?

- Lines 238--240: The phrase “the denominator vanishes’’ is ambiguous; it is unclear which denominator is being referred to. Clarifying this would make the subsequent discussion of instability easier to follow.

- Lines 246--249: The statement that increasing $\frac{1}{1-\kappa}$ requires "a finer step size $\eta$ and more inner steps $M$ to maintain the same approximation accuracy'' is not immediately obvious. Providing a short derivation or intuition based on the discretization bound in Theorem 3.6 would improve readability.

Overall, while the framework is interesting, the paper currently provides limited new theoretical insight or actionable guidance beyond existing analyses of adversarial or bilevel optimization. The analytical tools used in the paper—continuous-time ODE modeling and Lyapunov stability analysis—are well established in the study of optimization algorithms and min–max dynamics. While applying these tools to SAM is interesting, it is not obvious what fundamentally new insight this analysis provides beyond existing theoretical frameworks for bilevel or adversarial optimization. Therefore, while the paper introduces an interesting perspective on SAM dynamics, the practical implications and novelty relative to existing optimization theory remain somewhat limited.

# Minor Weaknesses
1. **Lack of justification for the ODE recovering SAM's objective.**
A foundational assertion (Lines 172–174) states that as $\beta/\alpha \to \infty$, the proposed two-timescale ODE (9) recovers the ``idealized ODE'' (4). This claim positions the ODE as the continuous-time analogue of SAM, yet no formal proof is provided in the text or appendix. While this may be intuitive to specialists, it is not self-evident. In particular, it would help to clarify why the inner flow:
    $$
    \frac{d\Delta}{dt} = \beta P_{\Delta}^{\perp}\left(\rho\frac{\nabla l(\theta+\Delta)}{\|\nabla l(\theta+\Delta) \|_2}-\Delta \right)
    $$
converges to a local maximizer $\Delta^*(\theta)$ of $\ell(\theta + \cdot)$ within the $\rho$-ball as $t \to \infty$.

2. **Clarity of the definition of $\kappa$ (division by zero).**
In Lines 221--223, the authors define
$$ c = \inf_{w \in \mathbb{R}^d} \|\nabla \ell(w)\|, \qquad \kappa := \rho \beta / c $$
(or a similar form). However, for most realistic loss functions in machine learning, stationary points exist where $\nabla \ell(w) = 0$, so $c = 0$ and $\kappa$ becomes undefined or infinite. Clarifying this point explicitly would help avoid potential confusion.

---

> ### Author Rebuttal · Authors · 2026-03-31
>
> **Q1: Unclear connection between ODE dynamics and SAM's preference for flat minima.**
>
> **Response:** The convergence set Ω automatically excludes high-curvature regions. Near a sharp minimum ($\nabla l(θ)=0$), the worst-case perturbation Δ aligns with the top Hessian eigenvector, lower-bounding the perturbed gradient by local sharpness: $||\nabla l(θ+Δ)||_2\approx ρ\sigma_1(\nabla^2 l(θ))$.
>
> A large $\sigma_1$ creates an exceedingly large perturbed gradient, violating Ω's equilibrium conditions and triggering a positive Lyapunov derivative that repels the trajectory. Consequently, the ODE cannot remain stationary in a sharp minimum and must escape. We will add a formal corollary highlighting this mechanism.
>
> **Q2: Claim regarding large M and contradiction with Theorem 3.6.**
>
> **Response:** We agree our wording around Lines 198-205 was stronger than Theorem 3.3 implies. Theorem 3.3 does not exclude all small-gradient stationary points for large β/α. Rather, its asymptotic convergence region contains an O(ρ) term growing with β/α from the residual threshold in the Lyapunov characterization. This reveals a trade-off: larger β/α improves inner-tracking fidelity but reduces convergence precision. Thus, increasing M is not guaranteed to improve final performance, despite improving approximation to the idealized ODE. We will revise the paper to state this carefully.
>
> Regarding the tension with Theorem 3.6, this theorem establishes an approximation fidelity, not a monotonic improvement guarantee. Increasing M makes discrete MSAM iterations better track the continuous-time dynamics. However, tracking the ODE closer doesn't inherently imply better final convergence because the ODE exhibits a trade-off controlled by β/α. Thus, increasing M improves tracking fidelity, but this does not guarantee better final optimization performance. We will explicitly clarify this distinction in the revision.
>
> **Q3: ρ not reported in Fig 1, gradual increase of M.**
>
> **Response:** Figure 1 uses ρ=1 (matching standard SAM), which we will explicitly state. To further illustrate Theorem 3.6, we re-run the experiment on the 2D synthetic landscape with intermediate M values using ρ=0.6, revealing a clear phase transition:
> * M=1: Trajectory sticks at saddle.
> * M=2,3: Escapes the saddle but converges to a sub-optimal local minimum (like GD).
> * M>=4: Successfully navigates the complex landscape to the true global minimum.
>
> This experiment is not intended to show that larger M is universally better, but rather to exhibit a regime in which insufficient inner accuracy leads to qualitatively wrong dynamics, and where increasing M beyond a threshold recovers the intended minimax dynamics and reaches the better solution.
>
> We also have the following Matrix Sensing Results:
>
> [rho = 0.05 Table](https://anonymous.4open.science/r/submission-results-630D/rho_0.05_table.md)
>
> [rho = 0.06 Table](https://anonymous.4open.science/r/submission-results-630D/rho_0.06_table.md)
>
> **Q4: Readability and clarity issues.**
>
> **Response:** We will clarify these points in the revision:
> * Transient Instability: Linked to a positive Lyapunov derivative forcing the trajectory out of high-curvature regions.
> * "Denominator vanishes": Replaced with: "the term 1-κ approaches zero, causing error bounds to diverge."
> * Scaling of η and M: Derivation added showing M must increase logarithmically for constant fixed-point error. As discretization error scales with 1-κ, η must decrease proportionally.
>
> **Q5: Lack of formal proof for ODE recovering SAM's objective.**
>
> **Response:** We will add a rigorous Appendix section proving:
> * Recovery of Idealized ODE: Using Tikhonov's Theorem, the fast variable $Δ\to Δ^*(θ)$, reducing slow dynamics to the idealized KKT conditions.
> * Inner Flow Convergence: Evaluating the loss's time derivative along the fast subsystem trajectory shows the projected gradient flow strictly increases the objective on the ρ-sphere until a local maximizer is reached.
>
> **Q6: Division by zero.**
>
> **Response:** We thank the reviewer for pointing this out. The singularity is inherent to normalized SAM: because the perturbation direction is normalized by a gradient norm, division-by-zero can arise near stationary points. In our notation, $w=θ+Δ$. Outside the convergence region Ω, we have by definition $||\nabla \ell(w)||>\max\\{Cρ,ρ\sigma_1(\nabla^2\ell(w))\\}\ge Cρ$, so the shifted gradient norm is uniformly bounded away from zero, and the corresponding infimum is strictly positive. We will revise the statement to make explicit that this lower bound is intended on the nonstationary regime outside Ω, rather than globally over all $w\in\mathbb{R}^d$.
>
> Near stationary points, the singularity is a known feature of normalized SAM rather than a flaw specific to our analysis. In practice, standard implementations use ε-smoothing, replacing $\|\nabla\ell(w)\|$ by $\|\nabla\ell(w)\|+ε$, which removes the division-by-zero issue. We will add a remark clarifying this point.

---

> > ### Author Rebuttal · Reviewer_xSYL · 2026-04-01
> >
> > Thank you for the detailed rebuttal and for addressing my concerns point by point.
> > The responses in Q3–Q6 satisfactorily resolve my concerns regarding the experimental evaluation, readability, and minor issues.
> >
> > However, I remain somewhat concerned about Q1 and Q2. While the intuitions provided are appealing and the underlying logic appears correct, the explanations are still relatively informal. Given that the main novelty of the paper lies in its theoretical analysis of SAM dynamics and the mechanistic insights derived from the ODE model, these sections would benefit from a more rigorous treatment. In particular, delivering the promised formal corollary in Q1 and carefully qualifying the trade-off argument in Q2 (including clearer statements of what Theorem 3.3 does and does not imply) would significantly strengthen the theoretical contributions.
> >
> > That said, I appreciate the authors' thoughtful reframing in Q2. Because the core intuition seems sound, I am willing to raise my overall score modestly in recognition of the rebuttal's improvements.

---

### Official Review · Reviewer_aySw · 2026-03-12

**Soundness:** 3
**Presentation:** 2
**Significance:** 4
**Originality:** 3
**Overall Recommendation:** 5
**Confidence:** 3

**Summary:**

This paper revisits the training dynamics of SAM through the lens of a continuous-time dynamical system. By studying the original minimax problem that inspires SAM, the author modeled a coupled ODE system for both the perturbation and the model parameter. A Lyapunov argument, free of convexity assumptions and valid for large radius, is applied to certify the stability and characterize the convergence set. This reveals the interesting orbit behavior around the optimal solution, and in high-curvature regions, the orbit bounces off and escapes the sharp minima. The discretization of the continuous system with explicit timescale separation naturally gives a multi-step variant of SAM, which iteratively solves a fixed-point equation in the inner loop to find an optimal perturbation, with approximation error contracting exponentially, and then updates based on the perturbed gradient. A Taylor expansion then connects the SAM update to the Penalized Gradient Norm objective in the small-ρ regime, framing SAM as an implicit gradient norm regularizer. Experiments on synthetic landscapes, matrix completion, and CIFAR validate the predicted role of ρ and M.

**Compliance With Llm Reviewing Policy:**

Affirmed.

**Final Justification:**

My concerns are adequately addressed.

**Key Questions For Authors:**

See weaknesses.

**Limitations:**

Not thoroughly discussed.

**Strengths And Weaknesses:**

Strengths:
1. Modeling the idealized dynamics for the min-max objective continuously as a two-timescale ODE is interesting and novel.
2. The theoretical discovery provides a unique perspective to understand SAM and how the perturbation radius, curvature, and gradient interplay during optimization. The framing of SAM's oscillation or orbital behavior is conceptually clean.
3. The timescale ratio elegantly bridges the continuous and discrete versions of the system and translates cleanly into practical guidance for the induced algorithm.
4. The failure case analysis for the contraction ratio gives practitioners a quantitative way of diagnosing when SAM struggles.
5. Importantly, the Lyapunov argument works without the convexity assumption, PL condition, or vanishingly small perturbation radius. This is a solid improvement upon prior research.

Weaknesses:
1. Section 3.5 seems a bit detached from the prior discussions, and the Taylor expansion argument sort of implicitly requires a small perturbation, which is a bit contradictory, as previous analysis works for a large radius.
2. The induced algorithm is conceptually interesting but problematic when it comes to practice. Each fixed point update requires a full gradient evaluation. In practice, SAM is already quite prohibitive due to the doubled gradient evaluation: extending it to $M$ steps only makes it worse.
3. Experiments are weak and poorly presented: the synthetic landscape experiment is an extreme simplification and can only be interpreted as a proof of concept; the matrix completion results are shown in 6 distinct tables and are rather confusing; CIFAR experiments show marginal improvements with no error statistics. Instead of trying to prove the algorithm's effectiveness, this paper may benefit more from experiments that validate the theoretical findings.
4. Presentation: The paper could use another walk-through to fix the minor presentation issues, like misnumbered proofs in Appendix A and B, repeated "see Appendix D for proof", Eq.3 uses capital $L$ while the rest uses $l$ for loss, negative sign dropped at Eq. 4, etc. Minor issues, but deserve attention.

---

> ### Author Rebuttal · Authors · 2026-03-31
>
> We sincerely thank the reviewer for recognizing the novelty of our two-timescale ODE modeling and the strength of our convexity-free Lyapunov arguments.
>
> **Q1: Section 3.5 seems detached, and the Taylor expansion implicitly assumes a small perturbation radius.**
>
> **Response:** We thank the reviewer for this helpful observation. We agree that the Taylor expansion in Section 3.5 implicitly corresponds to a small-$\rho$ regime. We would like to clarify, however, that Section 3.5 is not the foundation of our main theoretical results. The Lyapunov-based ODE stability result (Theorem 3.3) does not rely on this Taylor expansion, and the MSAM approximation result (Theorem 3.6) is also derived independently of Section 3.5.
>
> Our intention in Section 3.5 is to provide additional local interpretation and to connect our bilevel dynamical viewpoint with the penalized-gradient-norm perspective. The PGN-style ideas have been explored empirically in prior work (e.g., Zhao et al., 2022), and we show that, in the local regime where the Taylor expansion is appropriate, the continuous-time SAM dynamics naturally induce a PGN-like update through the curvature structure of the objective. We will revise Section 3.5 to clarify that this subsection is intended as a unifying interpretation rather than as a prerequisite for Theorems 3.3 and 3.6.
>
> **Q2: Computational overhead of the induced algorithm.**
>
> **Response:** We agree that requiring $M$ full gradient evaluations per step makes the naive MSAM implementation computationally heavy.To address the computational overhead, we introduce the **periodic MSAM** heuristic: We train ResNet-20 using standard SAM (\rho=0.05) for the first 80% of the training epochs. During the final 20% of the epochs (epochs 160-200), where the landscape becomes highly non-convex and the optimizer is more likely to encounter spurious local minima, we dynamically switch to MSAM with a larger perturbation radius (\rho=0.2) and multiple inner steps ($M$) to accurately resolve the worst-case landscape. The results and wall-clock training time can be found in the response to reviewer u1Wo Q3.
>
> **Q3: Weak experiments, 6 tables, synthetic landscape, no error statistics.**
>
> **Response:** We agree that the empirical presentation needed strengthening, and we will comprehensively update the experiments section:
> * **Error Statistics:**
> We have conducted multi-seeds experiments on the CIFAR-100 dataset using the ResNet-20 architecture on partial configurations due to the limited time of the rebuttal period. Each configuration was executed across five independent runs using distinct random seeds. All models were trained using a standard hyperparameter configuration: a fixed batch size of 128, a base learning rate of 0.1, a momentum of 0.9, and a weight decay of $5 \times 10^{-4}$. The final performance metrics are reported as the mean Top-1 validation accuracy alongside the standard deviation across these five runs, providing a statistically robust comparison between standard SGD, SAM, and our MSAM approach.
>
>   | Architecture | Optimizer | Perturbation Radius (ρ) | M Steps | Top-1 Accuracy (%) |
>   | :--- | :--- | :--- | :--- | :--- |
>   | ResNet-20 | SGD | 0.00 | 0 | 68.65 ± 0.41 |
>   | ResNet-20 | SAM | 0.05 | 0 | 69.46 ± 0.32 |
>   | ResNet-20 | MSAM | 0.10 | 5 | 70.01 ± 0.29 |
>
>   We refer to Appendix C.6 of the original SAM paper (Foret et al., 2020) for error statistics for WideResNet.In their ablation study ($\rho =0.05$) for both SAM and MSAM, evaluating a WideResNet architecture on the CIFAR-100 dataset, they demonstrated that increasing the inner steps ($M$) uncovers a significantly larger worst-case perturbation (captured by the "Estimated sharpness" metric).
>
>   | Inner Steps ($M$) | Test Accuracy (%) | Estimated Sharpness |
>   | :--- | :--- | :--- |
>   | 1 | 83.28 ± 0.08 | 0.82 ± 0.05 |
>   | 2 | 83.41 ± 0.08 | 1.83 ± 0.05 |
>   | 3 | 83.38 ± 0.09 | 2.36 ± 0.03 |
>   | 5 | 83.40 ± 0.06 | 2.82 ± 0.04 |
>
> * **Unified Matrix Completion:** We have repeated the matrix completion experiments with $11$ random seeds and will consolidate the results into a unified table along with error statistics in revision.
> * **Real-World Landscapes:** To bridge the gap between the synthetic 2D landscape and practical deep learning, we will add real-world 2D loss contour plots, visually confirming that MSAM locates wider, flatter minima.
>
> **Q4: Presentation issues (misnumbered proofs, Eq 3, Eq 4).**
>
> **Response:** We sincerely thank you for your meticulous reading. We have corrected these issues in the revision:
> * Fixed the misnumbered proofs in Appendices A and B.
> * Removed the duplicated references to Appendix D.
> * Added the missing minus sign in Eq. (4).
> * Clarified the notation for Eq. (3) by explicitly defining in the text that $L$ denotes the outer value function (the minimax objective), while $l$ denotes the base loss, resolving the notational inconsistency.

---

> > ### Author Rebuttal · Reviewer_aySw · 2026-03-31
> >
> > Thank you for the comprehensive response. My concerns are largely resolved, and I will update my score accordingly.

---

### Official Review · Reviewer_5gXj · 2026-03-13

**Soundness:** 2
**Presentation:** 2
**Significance:** 2
**Originality:** 2
**Overall Recommendation:** 4
**Confidence:** 4

**Summary:**

This paper studies the dynamics of SAM from a bilevel minimax perspective. It analyzes the optimality conditions of the inner maximization problem and, using Danskin’s theorem, shows that the derivative of the optimal perturbation can be ignored when computing the gradient. The authors further derive a two-timescale ODE that models the optimization trajectory of SAM as a coupled pursuit-game system. In addition, the paper presents MSAM as a natural discretization of this dynamical system, where performing multiple inner optimization steps reduces the approximation error. Finally, the authors conduct several experiments to validate their claims.

**Compliance With Llm Reviewing Policy:**

Affirmed.

**Final Justification:**

The authors’ response has addressed some of my concerns, but the limitations I pointed out still hold. Therefore, I will increase my score to 4.

**Key Questions For Authors:**

1. Could the authors clarify why it is desirable for the discrete updates to more closely follow the ODE trajectory? Are there scenarios in which this ODE can be considered optimal?

2. Does MSAM require roughly $m$ times additional backward passes to compute the perturbations? Have the authors considered ways to reduce this extra computational overhead?

3. The largest experiment in the paper is conducted on WRN28-2 with the CIFAR dataset. Could the authors provide results on larger-scale settings, for example at least on ImageNet?

**Limitations:**

yes

**Strengths And Weaknesses:**

**Pros:**
The theoretical analysis in this paper is generally sound. The discussion on the KKT conditions and the idealized dynamics is thorough, and the introduction of the two-timescale ODE is both natural and rigorous, providing some interesting insights for the community. Overall, the presentation is clear and easy to follow, and the claims are supported by multiple experiments, including both synthetic experiments and experiments on real-world datasets.

**Cons:**
1. Although I agree with most of the insights in the theoretical part, I believe the main weakness of this paper lies in its motivation. The primary motivation is to analyze the gap between SAM and the ideal minimax formulation and to reduce this gap via multi-step inner updates. However, the reason for wanting the discrete updates to more closely follow the ODE trajectory is not clearly justified. For example, using a smaller learning rate would also reduce the approximation error, but in practice overly small learning rates often significantly hurt generalization. I understand that the authors aim to focus more on convergence and optimization trajectories rather than generalization, but the current theory still struggles to explain both the practical successes and failures of SAM.

2. Although the paper includes multiple experiments, I did not find any particularly striking results. For instance, in Tables 7, 8, and 9, different values of $m$ do not appear to lead to significant performance differences.

3. The ideas of multi-step SAM and Penalized Gradient Norm discussed in this paper have already been mentioned in prior literature, so the novelty of these aspects is somewhat limited.

---

> ### Author Rebuttal · Authors · 2026-03-31
>
> **Q1: Could the authors clarify why it is desirable for the discrete updates to more closely follow the ODE trajectory? Are there scenarios in which this ODE can be considered optimal?**
>
> **Resoponse:** We agree this should be explained more clearly. Our motivation is not that following the ODE is desirable for its own sake. Rather, the ODE is the continuous-time representation of the bilevel minimax dynamics underlying SAM. It makes the coupled descent-ascent structure explicit and isolates how error from an inexact inner perturbation propagates into the outer update. Thus, improving ODE tracking is meant to better approximate the same minimax objective, not merely to reduce discretization error in general. This is also why a smaller outer learning rate is not an equivalent substitute: it changes the outer discretization, whereas improving the inner solve reduces the mismatch in the perturbation approximating the inner maximizer. We do not claim the ODE is ``optimal'' in a broad practical or generalization sense; it is the canonical continuous-time limit of the ideal minimax dynamics, and thus a natural reference for analyzing whether discrete SAM updates faithfully optimize the intended objective. More broadly, our theory targets optimization dynamics rather than a full explanation of SAM's generalization behavior. Questions about whether and when flatter solutions improve generalization remain actively debated in the literature, as acknowledged in Section 1.1 of our paper.
>
> **Q2:Does MSAM require roughly
>  times additional backward passes to compute the perturbations? Have the authors considered ways to reduce this extra computational overhead?**
>
> **Response**: A straightforward implementation of MSAM with
> M inner steps requires roughly M times the inner gradient computations. This is the tradeoff for reducing the perturbation approximation error, so a derived heuristic to overcome the computational overhead is to selectively use large M in regimes where the perturbation radius is larger or the local geometry is more challenging. Recognizing that standard SAM is sufficient for the simpler, early stages of optimization, we implemented the **periodic MSAM**. The detailed experiment setting, test accuracy and wall-clock time results can be found in our response to reviewer u1Wo Q3.
>
> **Q3: Could the authors provide results on larger-scale settings, for example at least on ImageNet?**
>
> **Response:** To address scalability, we evaluated Periodic MSAM against SAM on ResNet-50 over the Imagenette2 dataset. Using the same standard training recipe, baseline SAM $(\rho=0.1)$ achieved $75.01\%$ top-1 validation accuracy, while Periodic MSAM achieved $76.87\%$, a $+1.86\%$ gain. The Periodic MSAM model used standard SAM for the first $80\%$ of training and then switched to a multi-step phase $(\rho=0.2, M=5)$ for the final $20\%$. This suggests that a heavier multi-step phase can also be beneficial in a larger-scale vision setting.
>
> **Cons(2): Although the paper includes multiple experiments, I did not find any particularly striking results. For instance, in Tables 7, 8, and 9, different values of do not appear to lead to significant performance differences.**
>
> **Response:** We agree that the CIFAR accuracy differences across $M$ are modest in several settings, and we will revise the wording to avoid overstating them. Our claim is not that larger $M$ always yields large gains in test accuracy, but that it improves inner approximation fidelity, with regime-dependent effects on final performance. This is clearer in the controlled experiments: in matrix completion with 11 seeds, increasing $M$ consistently lowers both final loss and variance. For example, at $\rho=0.40$, SAM gives $5.4559\times 10^{-1}\pm 5.8210\times 10^{-1}$, while mSAM with $M=10$ gives $1.3844\times 10^{-4}\pm 6.1773\times 10^{-5}$. We will revise the presentation to emphasize that the synthetic and matrix completion experiments validate the dynamical mechanism, while CIFAR mainly serves as a practical sanity check.
>
>
> **Cons(3): The ideas of multi-step SAM and Penalized Gradient Norm discussed in this paper have already been mentioned in prior literature, so the novelty of these aspects is somewhat limited.**
>
> **Response:** We agree that ingredients such as multi-step inner updates and gradient-norm-related viewpoints have appeared before. Our novelty claim is not the first mention of these ideas in isolation, but the unified dynamical-systems interpretation: (i) formulating SAM explicitly as a bilevel minimax problem; (ii) deriving the coupled two-timescale ODE and characterizing its idealized dynamics; and (iii) showing how MSAM arises naturally as a discretization with reduced perturbation approximation error. We will revise the related work and contributions to distinguish more clearly between prior algorithmic ideas and our theoretical framing of why and when multiple inner steps improve approximation to the underlying minimax dynamics.

---

> > ### Author Rebuttal · Reviewer_5gXj · 2026-04-04
> >
> > Thank you for the authors’ response and the additional experiments. I will increase my score to 4.

---

### Official Review · Reviewer_u1Wo · 2026-03-15

**Soundness:** 3
**Presentation:** 3
**Significance:** 4
**Originality:** 4
**Overall Recommendation:** 5
**Confidence:** 3

**Summary:**

This paper introduces novel framework to analyze the behavior of sharpness-aware minimization (SAM), where SAM is understood via a continuous-time model of a coupling of parameter $\theta$ and perturbation $\Delta$. Such an understanding provides a solid ground for proposing Multiple-Step SAM (MSAM), which resolves the lack of interpretation of SAM behavior in avoiding sharp local minima. The number $M$ of inner iterations corresponding to one outer loop explicitly appears in the analysis and the analysis shows that whenever the region is with high curvature, (M-)SAM has a tendency to bounce off the local minimum.

**Compliance With Llm Reviewing Policy:**

Affirmed.

**Final Justification:**

My final decision for this paper is 'Accept'. This paper provides both thorough theoretical analysis and efficient method of improvement for Sharpness-Aware Minimization and the authors have adequately resolved my concerns.

**Key Questions For Authors:**

1. I might have misunderstood, but I wonder if there can be a case where the coupled $(\theta, \Delta)$-dynamics converge somewhere within the solution set $\Omega$, rather than oscillating in or orbiting the limit cycle within $\Omega$. (I am referencing Line 212-219, column 2.) It is true that it does not have a guarantee that it will converge to a single point, but there is also no guarantee that it will always oscillate as Theorem 3.3 is only about upper bound.

2. (This may align with my first question.) Merely out of curiosity, is there a possible case of existence of hallucinated minimizer, as illustrated in https://arxiv.org/abs/2509.21818? I guess additional inner loop update can resolve issue, but wonder how that can be explained in this framework.

3. Is there any strategy or heuristic to choose $\rho$ from the result of this paper? Or will $M$ and $\rho$ be a hyperparameter to be tuned?

**Limitations:**

Yes.

**Strengths And Weaknesses:**

This paper thoroughly analyzes both benefits and shortcomings of SAM through continuous-time model of coupling $(\theta, \Delta)$. Not only providing thorough analysis, this paper also comes up with a relatively simple solution called M-SAM with great intuitions in understanding the algorithm behavior. It was overall nicely organized from related concepts and existing problems to the proposed solution method.

These are minor comments:
1. Line 132, column 2: focus on the latter case of $\rightarrow$ focus on the **former** case of
2. Equation (4): Missing minus(-) sign on RHS
3. Line 194-195, column 2: Lyapunoc $\rightarrow$ **Lyapunov**
4. Theorem 3.3: Is the second term inside the $\max$ in RHS the multiple of $\sigma_1 (\cdot)$ by $\rho$? (Currently written as $\rho \vert \sigma_1 (\dots)$.)

---

> ### Author Rebuttal · Authors · 2026-03-31
>
> We deeply appreciate your positive assessment and meticulous review of our theoretical framework.
>
> **Q1: Convergence to a single point vs. limit cycle within $\Omega$.**
>
> **Response:** We appreciate this insightful question. We will add a formal equilibrium analysis in the revision. The key mechanism is the normalized perturbation $\Delta=\rho \frac{\nabla l}{\|\nabla l\|_2}$, whose norm is fixed at $\rho$. Unlike standard gradient descent, this normalized perturbation does not decay near a minimum, so the optimizer always evaluates the loss at a rigid $\rho$-distance from its current point. Thus, to converge to a strictly stationary single point would require $\nabla l(\theta+\Delta^*)=0$, which typically does not hold in a curved landscape when the gradient is evaluated at a fixed $\rho$-distance from the minimum. As a result, the trajectory enters a stable orbit where the outward normalized adversarial force is balanced by the inward pull of the loss landscape, creating a limit cycle within $\Omega$. To achieve true point-stationarity, an annealing schedule where $\rho \to 0$ is theoretically required.
>
> **Q2: Existence of hallucinated minimizers.**
>
> **Response:** Thank you for pointing out this highly relevant phenomenon. The existence of these spurious attractors—and their resolution via MSAM—is also intrinsically linked to the normalized gradient term.
>
> A hallucinated minimizer occurs in standard SAM when a single, rigid $\rho$-length normalized jump lands exactly on a flat spot, causing the perturbed gradient to vanish ($\nabla l(\theta + \Delta_{M=1}) = 0$). Because the step size is rigidly fixed by the normalization, $M=1$ lacks the flexibility to adjust and halts prematurely due to this 1-step truncation error.
>
> We will clarify in the revision that our method may not universally rule out hallucinated minimizers in pathological scenarios. For example, if the optimizer happens to initialize exactly at a point where $\nabla l(\theta + \Delta_{M=1}) = 0$, the gradient for the inner loop could also vanish, trapping the algorithm from the start.
>
> However, barring such exact initializations, taking multiple $\Delta$ updates structurally avoids these traps. By iteratively re-evaluating the normalized direction, the multiple inner steps allow tension and oscillation to build up within the $\rho$-neighborhood. This dynamic provides the necessary momentum to jump out of the hallucinated 1-step flat spots. As quantified in Theorem 3.6, employing MSAM with $M \ge 2$ exponentially reduces the truncation error by $O(\kappa^M)$. If the MSAM gradient vanishes outside of these initialization edge cases, our KKT analysis (Lemma 3.2) guarantees it is a true stationary point of the exact minimax objective.
>
> **Q3: Strategy or heuristic to choose $\rho$ and $M$.**
>
> **Response:** We appreciate the reviewer's question regarding how to choose $\rho$ and $M$ in practice. The choice of $\rho$ and $M$ is governed by a fundamental trade-off:
> * **Lower Bound (Theorem 3.6):** Approximation error scales as $O(\kappa^M)$. As $\rho$ increases, $M$ must also increase to keep truncation error bounded and avoid hallucinated minimizers.
> * **Upper Bound (Theorem 3.3):** Larger $M$ increases the effective timescale ratio $\beta/\alpha$, raising the stability threshold and the noise floor in the outer loop.
>
> We recommend a **periodic method** for practical use. Specifically, practitioners can apply a larger $M$ (e.g., $M \ge 3$) periodically to correct the trajectory and escape potential hallucinated minimizers, while using $M=1$ for intermediate steps to maintain computational efficiency. We have implemented this heuristic: in early stages of optimization, we train ResNet-20 using standard SAM ($\rho = 0.05$) for the first 80% of the training epochs. During the final 20% of the epochs (epochs 160-200), where the landscape becomes highly non-convex and the optimizer is more likely to encounter spurious local minima, we dynamically switch to MSAM with a larger perturbation radius ($\rho = 0.1, 0.2$) and multiple inner steps ($M$). As shown in the table below, Periodic MSAM consistently improves test accuracy while keeping the additional wall-clock training time highly manageable.
>
> | Method | Inner Steps ($M$) | $\rho$ (Last 20% Epochs) | Test Accuracy | Total Time |
> | :--- | :--- | :--- | :--- | :--- |
> | Standard SAM | 1 | 0.05 (All epochs) | 69.32% | 22m 5s |
> | Periodic MSAM | 5 | 0.2 | 69.76% | 27m 42s |
> | Periodic MSAM | 10 | 0.2 | 70.32% | 28m 1s |
> | Periodic MSAM | 10 | 0.1 | 70.01% | 31m 41s |
>
> **Q4: Minor comments (Typos, Eq 4, Lyapunov, Theorem 3.3 notation).**
>
> **Response:** We thank you for your meticulous reading. We will correct all typos in the revision: (1) Changed "latter case" to "former case" for Lemma 3.2. (2) Added the missing minus sign in Eq. (4). (3) Fixed "Lyapunov". (4) Clarified the notation in Theorem 3.3 to explicitly include $\rho$ multiplied by $\sigma_1$.

---

> > ### Author Rebuttal · Reviewer_u1Wo · 2026-04-04
> >
> > Dear authors, thank you for your thorough rebuttals.
> >
> > I believe all my concerns and key questions have been resolved adequately. I am keeping my score as is.

---

### Decision · Program_Chairs · 2026-04-30

**Decision:**

Accept (regular)

**Comment:**

This paper studies SAM through the lens of a minimax continuous-time dynamical system. By deriving a two-timescale ODE, the authors introduce a Lyapunov argument to analyze SAM's equilibrium and optimality gap. Furthermore, they propose a Multi-Step SAM (MSAM) algorithm that emerges from the discretization of this continuous flow. Initial reviews were generally positive, praising the theoretical soundness, the novel continuous-time interpretation, and the mathematically rigorous Lyapunov analysis. However, reviewers also raised several critiques such ascomputational overhead of MSAM, limited empirical validation (lacking large-scale experiments), and unclear justifications regarding how the ODE  prevents convergence to sharp minima.

The authors provided a comprehensive rebuttal that successfully swayed the reviewers. To mitigate the computational overhead and address the empirical weaknesses, the authors introduced a Periodic MSAM heuristic, provided new multi-seed error statistics, and included larger-scale evaluations. They also clarified the theory, supplied additional proofs, and committed to including a formal corollary linking the ODE dynamics to SAM's preference for flat minima. Following the discussion, Reviewers u1Wo and aySw considered their concerns fully resolved and maintained their strong scores. Reviewers 5gXj and xSYL acknowledged the substantial improvements (particularly the added experiments and the resolution of theoretical edge cases) and both increased their scores to a Weak Accept. While Reviewer xSYL noted that the explanation for flat minima preference remains slightly informal, and Reviewer 5gXj felt the algorithmic novelty was somewhat limited compared to prior literature, all reviewers ultimately agreed that the paper provides a valuable contribution.

Based on the entire review history, my final recommendation for this submission is Accept. The remaining critiques regarding algorithmic novelty are overshadowed by the paper's core theoretical contributions. The two-timescale ODE framework introduces an original and well-founded continuous-time perspective on SAM dynamics that the community is likely to build upon. The authors engaged constructively during the rebuttal period, successfully resolving the most critical empirical and theoretical gaps. Given the unified positive consensus among the reviewers and the theoretical foundation established in the work, this submission merits publication.